# Transitivity Recovering Decompositions: Interpretable and Robust Fine-Grained Relationships

**Abhra Chaudhuri**[1,5,6]    **Massimiliano Mancini**[2]    **Zeynep Akata**[3,4]    **Anjan Dutta**[5,6] *

[1] University of Exeter    [2] University of Trento    [3] University of Tübingen
[4] MPI for Informatics    [5] The Alan Turing Institute    [6] University of Surrey

## Abstract

Recent advances in fine-grained representation learning leverage local-to-global (emergent) relationships for achieving state-of-the-art results. The relational representations relied upon by such methods, however, are abstract. We aim to deconstruct this abstraction by expressing them as interpretable graphs over image views. We begin by theoretically showing that abstract relational representations are nothing but a way of recovering transitive relationships among local views. Based on this, we design Transitivity Recovering Decompositions (TRD), a graph-space search algorithm that identifies interpretable equivalents of abstract emergent relationships at both instance and class levels, and with no post-hoc computations. We additionally show that TRD is *provably robust* to noisy views, with empirical evidence also supporting this finding. The latter allows TRD to perform at par or even better than the state-of-the-art, while being fully interpretable. Implementation is available at `https://github.com/abhrac/trd`.

## 1  Introduction

Identifying discriminative object parts (local views) has traditionally served as a powerful approach for learning fine-grained representations [95, 85, 43]. Isolated local views, however, miss out on the larger, general structure of the object, and hence, need to be considered in conjunction with the global view for tasks like fine-grained visual categorization (FGVC) [94]. Additionally, the way in which local views combine to form the global view (local-to-global / emergent relationships [59]) has been identified as being crucial for distinguishing between classes that share the same set of parts, but differ only in the way the parts relate to each other [32, 97, 10, 9, 61, 12]. However, representations produced by state-of-the-art approaches that leverage the emergent relationships are encoded in an abstract fashion – for instance, through the summary embeddings of transformers [9, 10], or via aggregations on outputs from a GNN [97, 27]. This makes room for the following question that we aim to answer through this work: *how can we make such abstract relational representations more human-understandable?* Although there are several existing works that generate explanations for localized, fine-grained visual features [11, 78, 36, 63], providing interpretations for abstract representations obtained for discriminative, emergent relationships still remains unaddressed.

Illustrated in Figure 1, we propose to make existing relational representations interpretable through bypassing their abstractions, and expressing all computations in terms of a graph over image views, both at the level of the image as well as that of the class (termed *class proxy*, a generalized representation of a class). At the class level, this takes the form of what we call a Concept Graph: a generalized relational representation of the salient concepts across all instances of that class. We use graphs, as they are a naturally interpretable model of relational interactions for a variety of tasks [86, 18, 1, 46], allowing us to visualize entities (e.g., object parts, salient features), and their relationships.

---

*Abhra Chaudhuri (ac1151@exeter.ac.uk) is the corresponding author.

37th Conference on Neural Information Processing Systems (NeurIPS 2023).

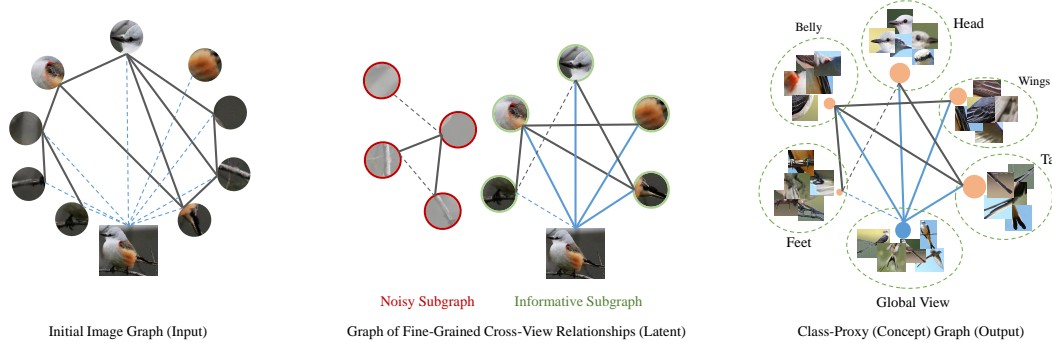

<p align="center">Initial Image Graph (Input)      Graph of Fine-Grained Cross-View Relationships (Latent)      Class-Proxy (Concept) Graph (Output)</p>

Figure 1: Instead of learning representations of emergent relationships that are abstract aggregations of views, our method deconstructs the input, latent, and the class representation (proxy) spaces into graphs, thereby ensuring that all stages along the inference path are interpretable.

We theoretically answer the posed question by introducing the notion of transitivity (Definition 3) for identifying *subsets* of local views that strongly influence the global view. We then show that the relational abstractions are nothing but a way of encoding transitivity, and design Transitivity Recovering Decompositions (TRD), an algorithm that decomposes both input images and output classes into graphs over views by recovering transitive cross-view relationships. Through the process of transitivity recovery (Section 3.2), TRD identifies maximally informative subgraphs that co-occur in the instance and the concept (class) graphs, providing end-to-end relational transparency. In practice, we implement TRD by optimizing a GNN to match a graph-based representation of the input image to the concept graph of its corresponding class (through minimizing the Hausdorff Edit Distance between the two). The input image graph is obtained by decomposing the image into its constituent views [10, 84, 94], and connecting complementary sets of views. The concept graph is obtained by performing an online clustering on the node and edge embeddings of the training instances of the corresponding class. The process is detailed in Section 3.3.

We also show that TRD is *provably robust* to noisy views in the input image (local views that do not / negatively contribute to the downstream task [13]). We perform careful empirical validations under various noise injection models to confirm that this is indeed the case in practice. The robustness allows TRD to always retain, and occasionally, even boost performance across small, medium, and large scale FGVC benchmarks, circumventing a known trade-off in the explainability literature [31, 73, 23, 37]. Finally, the TRD interpretations are ante-hoc – graphs encoding the learned relationships are produced as part of the inference process, requiring no post-hoc GNN explainers [88, 92, 34, 68, 90].

In summary, with the purpose of bringing interpretability to abstract relational representations, we make the following contributions – (1) show the existence of graphs that are equivalent to abstract local-to-global relational representations, and derive their information theoretic and topological properties; (2) design TRD, a provably robust algorithm that generates such graphs in an ante-hoc manner, incorporating transparency into the relationship computation pipeline at both instance and class levels; (3) extensive experiments demonstrating not only the achieved interpretability and robustness, but also state-of-the-art performance on benchmark FGVC datasets.

## 2 Related Work

**Fine-grained visual categorization:** Learning localized image features [2, 96, 47], with extensions exploiting the relationship between multiple images and between network layers [52] were shown to be foundational for FGVC. Strong inductive biases like normalized object poses [6, 87] and data-driven methods like deep metric learning [14] were used to tackle the high intra-class and low inter-class variations. Unsupervised part-based models leveraged CNN feature map activations [35, 94] or identified discriminative sequences of parts [4]. Novel ways of training CNNs for FGVC included boosting [55], kernel pooling [15], and channel masking [19]. Vision transformers, with their ability to learn localized features, had also shown great promise in FGVC [81, 29, 50]. FGVC interpretability has so far focused on the contribution of individual parts [11, 20]. However, the usefulness of emergent relationships for learning fine-grained visual features was demonstrated by

[83, 97, 9, 10]. Our method gets beyond the abstractions inherent in such approaches, and presents a fully transparent pipeline by expressing all computations in terms of graphs representing relationships, at both the instance and the class-level.

**Relation modeling in deep learning:** Relationships between entities serve as an important source of semantic information as has been demonstrated in graph representation learning [72, 18], deep reinforcement learning [93], question answering [67], object detection [33], knowledge distillation [60], and few-shot learning [70]. While the importance of learning relationships between different views of an image was demonstrated in the self-supervised context [61, 9], [10] showed how relational representations themselves can be leveraged for tasks like FGVC. However, existing works that leverage relational information for representation learning typically (1) model all possible ways the local views can combine to form the global view through a transformer, and distill out the information about the most optimal combination in the summary embedding [9, 10], or (2) model the underlying relations through a GNN, but performing an aggregation on its outputs [27, 97]. Both (1) and (2) produce vector-valued outputs, and such, cannot be decoded in a straightforward way to get an understanding of what the underlying emergent relationships between views are. This lack of transparency is what we refer to as "abstract". The above set of methods exhibit this abstraction not only at the instance, but at the class-level as well, which also appear as vector-valued embeddings. On the contrary, an interpretable relationship encoder should be able to produce graphs encoding relationships in the input, intermediate, and output spaces, while also representing a class as relationships among concepts, instead of single vectors that summarize information about emergence. This interpretability is precisely what we aim to achieve through this work.

**Explainability for Graph Neural Networks:** GNNExplainer [88] was the first framework that proposed explaining GNN predictions by identifying maximally informative subgraphs and subset of features that influence its predictions. This idea was extended in [92] through exploring subgraphs via Monte-Carlo tree search and identifying the most informative ones based on their Shapley values [69]. However, all such methods inherently provided either local [88, 34, 68] or global explanations [90], but not both. To address this issue, PGExplainer [51] presented a parameterized approach to provide generalized, class-level explanations for GNNs, while [82] obtained multi-grained explanations based on the pre-training (global) and fine-tuning (local) paradigm. [48] proposed a GNN explanation approach from a causal perspective, which led to better generalization and faster inference. On the other hand, [3] improved explanation robustness by modeling decision regions induced by similar graphs, while ensuring counterfactuality. Although ViG [27] presents an algorithm for representing images as a graph of its views, it exhibits an abstract computation pipeline lacking explainability. A further discussion on this is presented in Appendix A.10. Also, generating explanations for graph metric-spaces [99, 45] remains unexplored, which we aim to address through this work.

## 3 Transitivity Recovering Decompositions

Our methodology is designed around three central theorems - (i) Theorem 1 shows the existence of semantically equivalent graphs for every abstract representation of emergent relationships, (ii) Theorem 2 establishes an information theoretic criterion for identifying such a graph, and (iii) Theorem 3 shows how transitivity recovering functions can guide the search for candidate solutions that satisfy the above criterion. Additionally, Theorem 4 formalizes the robustness of transitivity recovering functions to noisy views. Due to space constraints, we defer all proofs to the Appendix A.9.

**Preliminaries:** Consider an image $\mathbf{x} \in \mathbb{X}$ with a categorical label $\mathbf{y} \in \mathbb{Y}$ from an FGVC task. Let $\mathbf{g} = c_g(\mathbf{x})$ and $\mathbb{L} = \{\mathbf{l}_1, \mathbf{l}_2, ... \mathbf{l}_k\} = c_l(\mathbf{x})$ be the global and set of local views of an image $\mathbf{x}$ respectively, jointly denoted as $\mathbb{V} = \{\mathbf{g}\} \cup \mathbb{L}$, where $c_g$ and $c_l$ are cropping functions applied on $\mathbf{x}$ to obtain such views. Let $f$ be a semantically consistent, relation-agnostic encoder (Appendices A.4 and A.7) that takes as input $\mathbf{v} \in \mathbb{V}$ and maps it to a latent space representation $\mathbf{z} \in \mathbb{R}^n$, where $n$ is the representation dimensionality. Specifically, the representations of the global view $\mathbf{g}$ and local views $\mathbb{L}$ obtained from $f$ are then denoted by $\mathbf{z}_g = f(\mathbf{g})$ and $\mathbb{Z}_\mathbb{L} = \{f(\mathbf{l}) : \mathbf{l} \in \mathbb{L}\} = \{\mathbf{z}_{l_1}, \mathbf{z}_{l_2}, ... \mathbf{z}_{l_k}\}$ respectively, together denoted as $\mathbb{Z}_\mathbb{V} = \{\mathbf{z}_g, \mathbf{z}_{l_1}, \mathbf{z}_{l_2}, ... \mathbf{z}_{l_k}\}$. Let $\xi$ be a function that encodes the relationships $\mathbf{r} \in \mathbb{R}^n$ between the global ($\mathbf{g}$) and the set of local ($\mathbb{L}$) views. DiNo [9] and Relational Proxies [10] can be considered as candidate implementations for $\xi$. Let $\mathbb{G}$ be a set of graphs $\{(\mathbb{V}, \mathbb{E}_1, \mathbb{F}_{\mathbb{V}_1}, \mathbb{F}_{\mathbb{E}_1}), (\mathbb{V}, \mathbb{E}_2, \mathbb{F}_{\mathbb{V}_2}, \mathbb{F}_{\mathbb{E}_2}), ...\}$, where the nodes in each graph constitute of the view set $\mathbb{V}$, and the edge set $\mathbb{E}_i \in \mathcal{P}(\mathbb{V} \times \mathbb{V})$, where $\mathcal{P}$ denotes the power set. $|\mathbb{G}| = |\mathcal{P}(\mathbb{V} \times \mathbb{V})|$, meaning

that $\mathbb{G}$ is the set of all possible graph topologies with $\mathbb{V}$ as the set of nodes. The node features $\mathbb{F}_{\mathbb{V}_i}$, and the edge features $\mathbb{F}_{\mathbb{E}_i} \in \mathbb{R}^n$.

## 3.1 Decomposing Relational Representations

**Definition 1** (**Semantic Relevance Graph**). A graph $\mathcal{G}_s \in \mathbb{G}$ of image views where each view-pair is connected by an edge with strength proportional to their joint relevance in forming the semantic label $\mathbf{y}$ of the image. Formally, the weight of an edge $E_{S_{ij}}$ connecting $\mathbf{v}_i$ and $\mathbf{v}_j$ is proportional to $I(\mathbf{v}_i\mathbf{v}_j; \mathbf{y})$.

Intuitively, $E_{S_{ij}}$ is a measure of how well the two views pair together as compatible puzzle pieces in depicting the central concept in the image.

**Theorem 1.** *For a relational representation* $\mathbf{r}$ *that minimizes the information gap* $I(\mathbf{x}; \mathbf{y}|\mathbb{Z}_{\mathbb{V}})$, *there exists a metric space* $(\mathcal{M}, d)$ *that defines a Semantic Relevance Graph such that:*

$$d(\mathbf{z}_i, \mathbf{y}) > \delta \wedge d(\mathbf{z}_j, \mathbf{y}) > \delta \Rightarrow$$

$$\exists\, \xi : \mathbb{R}^n \xrightarrow{sem} \mathbb{R}^n \mid \mathbf{r} = \xi(\varphi(\mathbf{z}_i), \varphi(\mathbf{z}_j)), d(\mathbf{r}, \mathbf{y}) \leq \delta$$

*where* $\mathcal{M} \in \mathbb{R}^n$, $\xrightarrow{sem}$ *and* $d$ *denote semantically consistent transformations and distance metrics respectively,* $\varphi : \mathbb{Z}_{\mathbb{V}} \xrightarrow{sem} \mathcal{M}$, *and* $\delta$ *is some finite, empirical bound on semantic distance.*

*Intuition*: The theorem says that, even if $\mathbf{z}_i$ and $\mathbf{z}_j$ individually cannot encode anymore semantic information, the metric space $(\mathcal{M}, d)$ encodes their relational semantics via its distance function $d$. Assuming the theorem to be false would mean that the outputs of the aggregation do not allow for a distance computation that is semantically meaningful, and hence, there is no source from which $\mathbf{r}$ (the relational embedding) can be learned. This would imply that either (i) the information gap (Appendix A.2) does not exist, which is false, because $f$ is a relation-agnostic encoder, or (ii) all available metric spaces are relation-agnostic and hence, the information gap would always persist. The latter is again in contrary to what is shown in Appendix A.6, which derives the necessary and sufficient conditions for a model to bridge the information gap. Thus, $(\mathcal{M}, d)$ defines $\mathcal{G}_s$ with weights $1/d(\varphi(\mathbf{v}_i), \varphi(\mathbf{v}_j)) \propto I(\mathbf{v}_i\mathbf{v}_j; \mathbf{y})$, where $\varphi : \mathbb{V} \to \mathcal{M}$.

**Corollary 1.1** (Proxy / Concept Graph). *Proxies can also be represented by graphs.*

*Intuition*: Since $(\mathcal{M}, d)$ is equipped with a semantically relevant distance function, the hypersphere enclosing a locality/cluster of local/global view node embeddings can be identified as a semantic relevance neighborhood. The centers of such hyperspheres can be considered as a proxy representation for each such view, generalizing some salient concept of a class. The centers of each such view cluster can then act as the nodes of a proxy/concept graph, connected by edges, that are also obtained in a similar manner by clustering the instance edge embeddings.

**Theorem 2.** *The graph* $\mathcal{G}^*$ *underlying* $\mathbf{r}$ *is a member of* $\mathbb{G}$ *with maximal label information. Formally,*

$$\mathcal{G}^* = \arg\max_{\mathcal{G} \in \mathbb{G}} I(\mathcal{G}, \mathbf{y})$$

*Intuition*: The label information $I(\mathbf{x}, \mathbf{y})$ can be factorized into relation-agnostic and relation-aware components (Appendix A.2). The node embeddings in $\mathcal{G}^*$ already capture the relation-agnostic component (being derived from $f$). From the implication in Theorem 1, we know that given the node embeddings, $\xi$ is able to reduce further uncertainty about the label by joint observation of view pairs (or in other words, edges in $\mathcal{G}^*$). Hence $\xi$ must be relation-aware, as the relation-agnostic component is already captured by $f$. Thus, $G^*$, which is obtained as a composition of $f$ and $\xi$, must be a *sufficient* representation of $\mathbf{x}$ (Appendices A.2 and A.3). Since sufficiency is the upper-bound on the amount of label information that a representation can encode (Appendix A.3), $\mathcal{G}^*$ is the graph that satisfies the condition in the theorem. In the section below, we show that the way to obtain $\mathcal{G}^*$ from $\mathbb{G}$ is by recovering transitive relationships among local views.

## 3.2 Transitivity Recovery

**Definition 2** (**Emergence**). The degree to which a set of local views $(\mathbf{v}_1, \mathbf{v}_2, ...\mathbf{v}_k)$ contributes towards forming the global view. It can be quantified as $I(\mathbf{v}_1\mathbf{v}_2...\mathbf{v}_k; \mathbf{g})$.

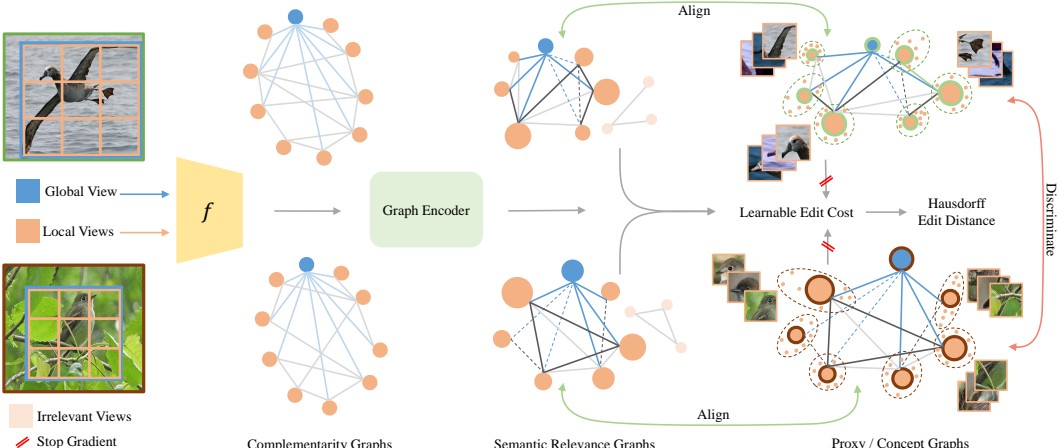

Figure 2: The TRD pipeline: We begin by computing a Complementarity Graph ($\mathcal{G}_c$) on the view embeddings obtained from $f$. The Graph Encoder derives the Semantic Relevance Graph ($\mathcal{G}_s$) through transitivity recovery on $\mathcal{G}_c$. We perform a class-level clustering of the instance node and edge embeddings, producing a proxy graph, representing the salient concepts of a class and their interrelationships. The sufficiency of $\mathcal{G}_s$ is guaranteed by minimizing its Hausdorff distance with its proxy graph.

**Definition 3** (**Transitivity**). Local views $\mathbf{v}_1, \mathbf{v}_2$, and $\mathbf{v}_3$ are transitively related *iff*, when any two view pairs have a high contribution towards emergence, the third pair also has a high contribution. Formally,

$$I(\mathbf{v}_i\mathbf{v}_j; \mathbf{g}) > \gamma \wedge I(\mathbf{v}_j\mathbf{v}_k; \mathbf{g}) > \gamma \Rightarrow I(\mathbf{v}_i\mathbf{v}_k; \mathbf{g}) > \gamma,$$

where $(i, j, k) \in \{1, 2, 3\}$ and $\gamma$ is some threshold for emergence. A function $\varphi(\mathbf{v})$ is said to be *transitivity recovering* if the transitivity between $\mathbf{v}_1, \mathbf{v}_2$, and $\mathbf{v}_3$ can, in some way, be inferred from the output space of $\varphi$. This helps us quantify the *transitivity* of *emergent* relationships across partially overlapping sets of views.

**Theorem 3.** *A function $\varphi(\mathbf{z})$ that reduces the uncertainty $I(\mathbf{v}_i\mathbf{v}_j; \mathbf{g}|\varphi(\mathbf{z}_i)\varphi(\mathbf{z}_j))$ about the emergent relationships among the set of views can define a $\mathcal{G}_s$ by being transitivity recovering.*

*Intuition*: Reducing the uncertainty about transitivity among the set of views is equivalent to reducing the uncertainty about the emergent relationships, and hence, bridging the information gap. If $\mathbf{v_1}, \mathbf{v_2}$, and $\mathbf{v_3}$ are not transitively related, then it would imply that, at most, only one of the three views ($\mathbf{v}_j$) is semantically relevant, and the other two ($\mathbf{v}_i, \mathbf{v}_k$) are not. This leads to a *degenerate* form of emergence. Thus, transitivity is the key property in the graph space that filters out such degenerate subsets in $\mathbb{L}$, helping identify view triplets where all the elements contribute towards emergence. A further discussion can be found in Appendix A.11. Now, we know that relational information must be leveraged to learn a sufficient representation. So, for sufficiency, a classifier operating in the graph space must leverage transitivity to match instances and proxies. In the proof, we hence show that transitivity in $\mathbb{G}$ is equivalent to the relational information $\mathbf{r} \in \mathbb{R}^n$. Thus, the explanation graph $\mathcal{G}^*$ is a maximal MI member of $\mathbb{G}$ obtained by applying a transitivity recovering transformation on the Complementarity Graph.

### 3.3 Generating Interpretable Relational Representations

Depicted in Figure 2, the core idea behind TRD is to ensure relational transparency by expressing all computations leading to classification on a graph of image views (Theorem 1), all the way up to the class-proxy, by decomposing the latter into a concept graph (Corollary 1.1). We ensure the sufficiency (Theorem 2) of the instance and proxy graphs through transitivity recovery (Theorem 3) by Hausdorff Edit Distance minimization.

**Complementarity Graph:** We start by obtaining relational-agnostic representations $\mathbb{Z}_\mathbb{V}$ from the input image $\mathbf{x}$ by encoding each of its views $\mathbf{v} \in \mathbb{V}$ as $f(\mathbf{v})$. We then obtain a Complementarity Graph $\mathcal{G}_c \in \mathbb{G}$ with node features $\mathbb{F}_\mathbb{V} = \mathbb{Z}_\mathbb{V}$, and edge strengths inversely proportional to the value of the mutual information between the local view embeddings $I(\mathbf{z}_i, \mathbf{z}_j)$ that the edge connects. Specifically, we instantiate the edge features $\mathbb{F}_\mathbb{E}$ as learnable $n$-dimensional vectors, all with the same value of

$1/|\mathbf{z}_i \cdot \mathbf{z}_j|$. This particular choice of calculating the edge weights is valid under the assumption that $f$ is a semantically consistent function. Intuitively, local views with low mutual information (MI) are complementary to each other, while ones with high MI have a lot of redundancy. The *inductive bias* behind the construction of such a graph is that, strengthening the connections among the complementary pairs suppresses the flow of redundant information during message passing, thereby reducing the search space in $\mathbb{G}$ for finding $\mathcal{G}^*$. The global view $\mathbf{g}$, however, is connected to all the local-views with a uniform edge weight of $\mathbb{1}^n$. This is because the local-to-global emergent information, *i.e.*, $I(\mathbf{l}_i\mathbf{l}_j...; \mathbf{g})$, is what we want the model to discover through the learning process, and hence, should not incorporate it as an inductive bias.

**Semantic Relevance Graph:** We compute the Semantic Relevance Graph $\mathcal{G}_s$ by propagating $\mathcal{G}_c$ through a Graph Attention Network (GAT) [77]. The node embeddings obtained from the GAT correspond to the $\varphi(\mathbf{z})$ in our theorems. We ensure that $\varphi$ is transitivity recovering by minimizing the Learnable Hausdorff Edit Distance between the instance and the proxy graphs (Corollary 1.1), which is known to take into account the degree of local-to-global transitivity for dissimilarity computation [64]. Below, we discuss this process in further detail.

**Proxy / Concept Graph:** We obtain the proxy / concept graph for each class via an online clustering of the Semantic Relevance node and edge embeddings ($\varphi_n(\mathbf{z})$, $\varphi_e(\mathbf{z})$ respectively) of the train-set instances of that class, using the Sinkhorn-Knopp algorithm [16, 8]. We set the number of node clusters to be equal to $|V|$. The number of edge clusters (connecting the node clusters) is equal to $|V|(|V|-1)/2$. Note that this is the case because all the graphs throughout our pipeline are complete. Based on our theory of transitivity recovery, sets of semantically relevant nodes should form cliques in the semantic relevance and proxy graphs, and remain disconnected from the irrelevant ones by learning an edge weight of $\mathbb{0}^n$. We denote the set of class proxy graphs by $\mathbb{P} = \{\mathcal{G}_{\mathbf{P}_1}, \mathcal{G}_{\mathbf{P}_2}, ..., \mathcal{G}_{\mathbf{P}_k}\}$, where $k$ is the number of classes.

**Inference and Learning objective:** To recover end-to-end explainability, we need to avoid converting the instance and the proxy graphs to abstract vector-valued embeddings in $\mathbb{R}^n$ at all times. For this purpose, we perform matching between $\mathcal{G}_s$ and $\mathcal{G}_{\mathbf{p}} \in \mathbb{P}$ via a graph kernel [22]. This kernel trick helps us bypass the computation of the abstract relationship $\xi : \mathbb{G} \to \mathbb{R}^{n\ 2}$, and perform the distance computation directly in $\mathbb{G}$, using only the graph-based decompositions, and thereby keeping the full pipeline end-to-end explainable.

We choose to use the Graph Edit Distance (GED) as our kernel [58]. Although computing GED has been proven to be NP-Complete [79], quadratic time Hausdorff Edit Distance (HED) [25] and learning-based approximations [65] have been shown to be of practical use. However, such approximations are rather coarse-grained, as they are either based on linearity assumptions [65], or consider only the substitution operations [25]. Learnable (L)-HED [64] introduced two additional learnable costs for node insertions and deletions using a GNN, thereby making it a more accurate approximation. We observe that our $\varphi$ already performs the job of the encoder prefix in L-HED (Appendix A.12). Thus, we simply apply a fully connected MLP head $\psi : \mathcal{M} \to \mathbb{R}$ with shared weights on the $\varphi(\mathbf{z})$ embeddings and the vertices of $\mathcal{G}_p \in \mathbb{P}$ to compute the node insertion and deletion costs. Thus, in our case, L-HED takes the following form:

$$h(\mathcal{G}_s, \mathcal{G}_p) = \alpha \left( \sum_{\mathbf{u} \in \mathcal{G}_s} \min_{\mathbf{v} \in \mathcal{G}_p} c(\mathbf{u}, \mathbf{v}) + \sum_{\mathbf{v} \in \mathcal{G}_p} \min_{\mathbf{u} \in \mathcal{G}_s} c(\mathbf{u}, \mathbf{v}) \right) \tag{1}$$

$$c(\mathbf{u}, \mathbf{v}) = \begin{cases} ||\psi(\mathbf{u})||, & \text{for deletions } (\mathbf{u} \to \varepsilon), \\ ||\psi(\mathbf{v})||, & \text{for insertions } (\varepsilon \to \mathbf{v}), \\ ||\mathbf{u} - \mathbf{v}||/2, & \text{for substitutions,} \end{cases}$$

where $\alpha = 1/(2|\mathbb{V}|)$, and $\varepsilon$ is the empty node [64]. We use $h(\cdot, \cdot)$ as a distance metric for the Proxy Anchor Loss [39], which forms our final learning objective, satisfying Theorem 2. Below we discuss how transitivity recovery additionally *guarantees* robustness to noisy views.

### 3.4 Robustness

Let $\mathcal{D}$ be the data distribution over $\mathbb{X} \times \mathbb{Y}$. Consider a sample $\mathbf{x} \in \mathbb{X}$ composed of views $\mathbb{V} = \{\mathbf{v}_1, \mathbf{v}_2, ..., \mathbf{v}_k\}$ with label $\mathbf{y} \in \mathbb{Y}$. A view $\mathbf{v} \in \mathbb{V}$ is said to be *noisy* if $\text{label}(\mathbf{v}) \neq y$ [13]. The

---

[2]Graph kernels, in general, help bypass mappings $\xi : \mathbb{G} \to \mathcal{H}$, where $\mathcal{H}$ is the Hilbert space [22].

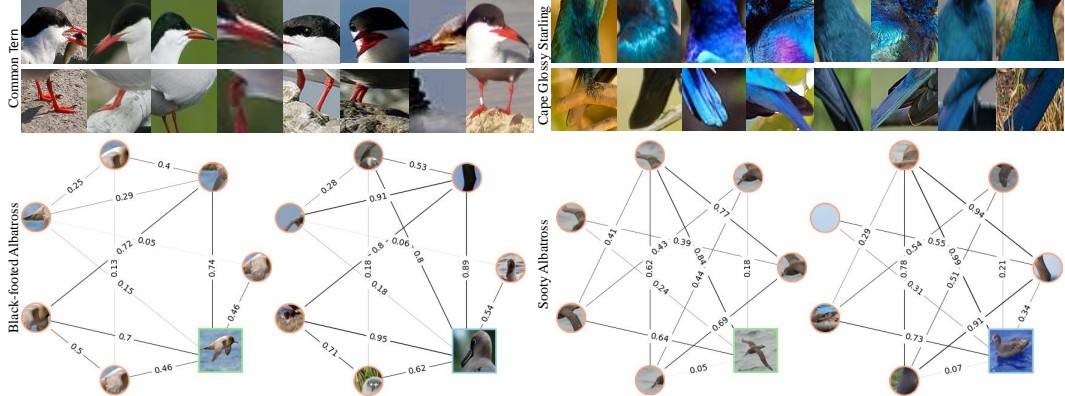

Figure 3: Sample TRD explanations on CUB. Top: 8 nearest neighbors (columns) of the top-2 proxy nodes / concepts (rows) with highest emergence (Definition 2). Bottom: Instance (global view with green border) and proxy (global view with blue border) explanation graphs constructed using the top-6 nodes (nearest train set neighbor for the proxy nodes) with highest emergence. The edge-weights are $L_2$-norms of the edge embeddings (best visible on screen with zooming).

fraction of noisy views for a given sample is denoted by $\eta$. Let $f$ be a classifier that minimizes the empirical risk of the mapping $f(\mathbf{x}) \mapsto \mathbf{y}$ in the ideal noise-free setting. Let $\mathbb{V}^\eta$ be the set of noisy views in $\mathbb{V}$ with topology $\tau^\eta$. Let $\mathbb{V}^* = \mathbb{V} - \mathbb{V}^\eta$ be the set of noise free (and hence, transitively related - Theorem 3) views with topology $\tau^*$. Recall that $\mathcal{G}^*$ is the optimal graph of $\mathbb{V}$ satisfying the sufficiency criterion in Theorem 2, and $\mathcal{G}_{\mathbf{p}} \in \mathbb{P}$ is its proxy graph.

**Theorem 4.** *In the representation space of $\varphi$, the uncertainty in estimating the topology $\tau^\eta$ of $\mathbb{V}^\eta$ is exponentially greater in the error rate $\epsilon$, than the uncertainty in estimating the topology $\tau^*$ of $\mathbb{V}^*$.*

*Intuition*: This theorem is based on the fact the number of topologies that a set of noisy views can assume, is exponentially more (in the desired error rate) to what can be assumed by a set of transitive views. Since $\varphi$ is explicitly designed to be transitivity recovering, making predictions using $\varphi$ based on noisy views would go directly against its optimization objective of reducing the output entropy (due to the exponentially larger family of topologies to choose from). So, to minimize the uncertainty in Equation (1), $\varphi$ would always make predictions based on transitive views, disregarding all noisy views in the process, satisfying the formal requirement of robustness (Definition 4). In other words, the following property of TRD allowed us to arrive at this result – the structural priors that we have on the transitive subgraphs make them the most optimal candidates for reducing the prediction uncertainty, relative to the isolated subgraphs of noisy views which do not exhibit such regularities.

## 4 Experiments

### 4.1 Experimental Settings & Datasets

**Implementation Details:** For obtaining the global view, we follow [84, 94] by selecting the smallest bounding box containing the largest connected component of the thresholded final layer feature map obtained from an ImageNet-1K [17] pre-trained ResNet50 [30], which we also use as the relation-agnostic encoder $f$. The global view is resized to $224 \times 224$. We then obtain 64 local views by randomly cropping $28 \times 28$ regions within the global crop and resizing them to $224 \times 224$. We use a 8-layer Graph Attention Network (GAT) ([77], with 4 attention heads in each hidden layer, and normalized via GraphNorm [7] to obtain the Semantic Relevance Graph. We train TRD for 1000 epochs using the Adam optimizer, at an initial learning rate of 0.005 (decayed by a factor of 0.1 every 100 epochs), with a weight decay of $5 \times 10^{-4}$. We generally followed [77, 10] for choosing the above settings. We implement TRD with a single NVIDIA GeForce RTX 3090 GPU, an 8-core Intel Xeon processor, and 32 GBs of RAM.

**Datasets:** To verify the generalizability and scalability of our method, we evaluate it on small, medium and large-scale FGVC benchmarks. We perform small-scale evaluation on the Soy and Cotton Cultivar datasets [89], while we choose FGVC Aircraft [53], Stanford Cars [42], CUB [80],

and NA Birds [75] for medium scale evaluation. For large-scale evaluation, we choose the iNaturalist dataset [76], which has over 675K train set and 182K test set images.

## 4.2  Interpretability & Robustness

**Qualitative Results:** Figure 3 shows sample explanations obtained using TRD on the CUB dataset. The top rows represent proxy nodes (concepts), and the columns represent the 8 nearest neighbors of the corresponding proxy nodes, which is highly consistent across concepts and classes. In the bottom are instance and proxy explanation graphs (top-6 nodes with highest emergence) from very similar-looking but different classes. For the proxies, the nearest train set neighbors to the node embeddings are visualized. Even with similar appearance, the graphs have very different structures. This shows that TRD is able to recover the discriminative relational properties of an image, and encode them as graphs. We provide additional visualizations in the supplementary.

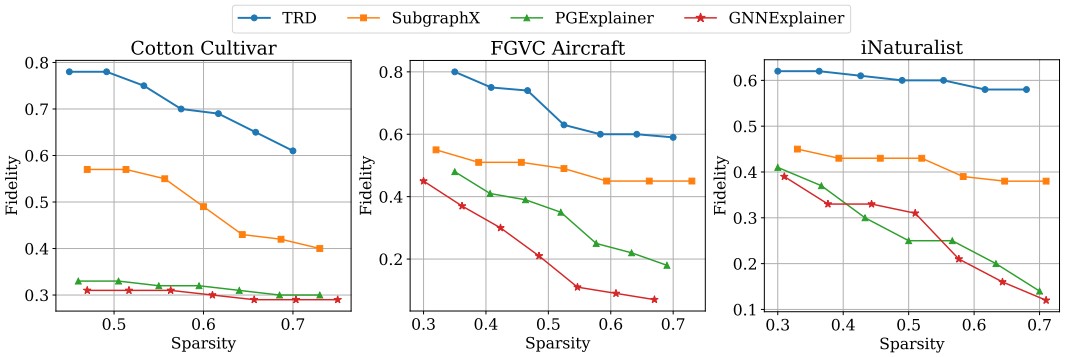

Figure 4: Fidelity *vs* Sparsity curves of TRD and SOTA GNN interpretability methods.

**Quantitative Evaluation:** We quantitatively evaluate the explanations obtained using TRD based on the Fidelity (relevance of the explanations to the downstream task) *vs* Sparsity (precision of the explanations) plots for the generated explanations, which is a standard way to measure the efficacy of GNN explainability algorithms [92, 62, 91]. We compare against SOTA, generic GNN explainers, namely SubgraphX [92], PGExplainer [51], and GNNExplainer [88], on candidate small (Cotton), medium (FGVC Aircraft), and large (iNaturalist) scale datasets, and report our findings in Figure 4. TRD consitently provides the highest levels of Fidelity not only in the dense, but also in the high sparsity regimes, outperforming generic GNN explainers by significant margins. Unlike the generic explainers, since TRD takes into account the edit distance with the proxy graph for computing the explanations, it is able to leverage the class-level topological information, thereby achieving higher precision. The supplementary contains results on the remaining datasets, details on the Fidelity and Sparsity metrics, and experiments on the functional equivalence of TRD with post-hoc explanations.

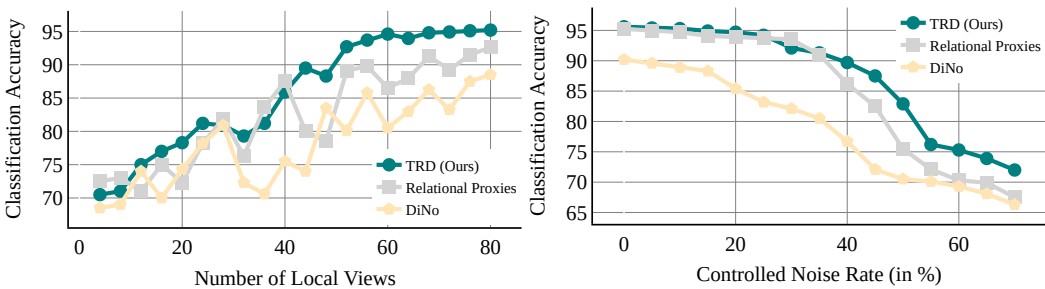

Figure 5: Classification accuracies of relational learning based methods with different noise models on FGVC Aircraft. Left: Increasing number of local views sampled randomly across the entire image (instead of just the global view). Right: Increasing the *proportion* of noisy views in a controlled manner by sampling from the region outside of the global view.

**Robustness to Noisy-Views:** Figure 5 shows the performance of TRD on FGVC Aircraft under the following two noise injection models – (1) The local views are sampled uniformly at random across

the entire image, instead of just the global view. This model thus randomizes the amount of label information present in $|\mathbb{L}|$. (2) A fraction $\eta$ of the local views are sampled from outside the global view, and the remaining ones $(1 - \eta)$ come from within the global view, where $\eta$ is the variable across experiments. This puts a fixed upper bound to the amount of label information in $|\mathbb{L}|$. It can be seen that under model (1), TRD is significantly more stable to changing degrees of uncertainty in label information compared to other relational learning-based methods. Under model (2), TRD outperforms existing methods as the amount of label information is decreased by deterministically increasing the noise rate. As discussed in Section 3.4 and formally proved in Appendix A.9.1, transitivity acts as a semantic invariant between the instance and the proxy graph spaces, which is a condition that subgraphs induced by the noisy views do not satisfy, leading to the observed robustness.

**Effect of Causal Interventions:** Here, we aim to understand the behaviour of our model under corruptions that affect the underlying causal factors of the data generating process. To this end, we train TRD by replacing a subset of the local views for each instance with local views from other classes, both during training and inference. As the proxies are obtained via a clustering of the instance graphs, these local views consequently influence the proxy graphs. We report our quantitative and qualitative findings in Table 1 and Figure 6 (Appendix A.17) respectively. TRD significantly outperforms Relational Proxies at all noise rates (percentage of local views replaced), and the gap between their performances widens as the percentage of corruption increases (Table 1). Qualitatively (Figure 6), our model successfully disregards the views introduced from the negative class at both the instance and proxy level. Such views can be seen as being very weakly connected to the global view, as well as the correct set of local views that actually belong to that class. Under this causal intervention, the TRD objective is thus equivalent to performing classification while having access to only the subgraph of clean views from the correct class.

| $\eta$ | 10 | 20 | 30 | 40 | 50 |
|---|---|---|---|---|---|
| Relational Proxies | 93.22 | 87.12 | 79.35 | 70.99 | 63.60 |
| **TRD (Ours)** | **94.90** | **91.54** | **82.80** | **76.35** | **70.55** |

Table 1: Quantitative performance comparison between Relational Proxies and TRD with increasing rates of corruption ($\eta$: percentage local views from different classes).

## 4.3 FGVC Performance

**Comparison with SOTA:** Table 2 compares the performance of TRD with SOTA on benchmark FGVC datasets. Some of the most commonly used approaches involve some form of regularizer [21, 41], bilinear features [98], or transformers [29, 81]. However, methods that use multi-view information [94, 49], especially their relationships [4, 97, 10, 5, 71] for prediction have been the most promising, although their final image and class representations are abstract. TRD can be seen to provide the best performance across all benchmarks, while also being fully interpretable. We attribute this to the robustness of our method to noisy views, which is known to have a significant impact on metric learning algorithms [13].

**Ablation Studies:** Table 3 shows the contributions of the components of TRD, namely Complementarity Graph: CG (Section 3.3), Proxy Decomposition: PD (Corollary 1.1), and Transitivity Recovery: TR (Theorem 3) to its downstream classification accuracy on the FGVC Aircraft, Cotton, and Soy Cultivar datasets. Row 0 is the baseline with none of our novel components. Row 1 corresponds to the setting where the proxy representation is still abstract, *i.e.*, in $\mathbb{R}^n$. We concatenate the nodes in $\mathcal{G}_s$ and propagate it through a 1-hidden layer MLP to summarize it in $\mathbb{R}^n$. As we decompose the proxy into a graph (PD, Row 2) based on the instance node and edge clustering-based approach as discussed in Section 3.3 for class-level interpretability, we can see that the performance

| ID | CG | PD | TR | Aircraft | Cotton | Soy |
|---|---|---|---|---|---|---|
| 0. | | | | 94.60 | 67.70 | 46.00 |
| 1. | ✓ | | | 95.10 | 69.60 | 47.70 |
| 2. | | ✓ | | 94.94 | 68.31 | 46.70 |
| 3. | ✓ | ✓ | | 95.15 | 69.70 | 50.32 |
| 4. | | ✓ | ✓ | 95.40 | 70.10 | 50.99 |
| 5. | ✓ | ✓ | ✓ | **95.60** | **70.90** | **52.15** |

Table 3: Ablations on the key components of TRD namely Complementarity Graph (CG), Proxy Decomposition (PD), and TR (Transitivity Recovery).

| Method | Small | | Medium | | | | Large |
|---|---|---|---|---|---|---|---|
| | Cotton | Soy | FGVC Aircraft | Stanford Cars | CUB | NA Birds | iNaturalist |
| MaxEnt, NeurIPS'18 | - | - | 89.76 | 93.85 | 86.54 | - | - |
| DBTNet, NeurIPS'19 | - | - | 91.60 | 94.50 | 88.10 | - | - |
| StochNorm, NeurIPS'20 | 45.41 | 38.50 | 81.79 | 87.57 | 79.71 | 74.94 | 60.75 |
| MMAL, MMM'21 | 65.00 | 47.00 | 94.70 | 95.00 | 89.60 | 87.10 | 69.85 |
| FFVT, BMVC'21 | 57.92 | 44.17 | 79.80 | 91.25 | 91.65 | 89.42 | 70.30 |
| CAP, AAAI'21 | - | - | 94.90 | 95.70 | 91.80 | 91.00 | - |
| GaRD, CVPR'21 | 64.80 | 47.35 | 94.30 | 95.10 | 89.60 | 88.00 | 69.90 |
| WTFocus, ECCV'22 | - | - | 94.70 | 95.30 | 90.80 | - | - |
| SR-GNN, TIP'22 | - | - | 95.40 | 96.10 | 91.90 | - | - |
| TransFG, AAAI'22 | 45.84 | 38.67 | 80.59 | 94.80 | 91.70 | 90.80 | 71.70 |
| Relational Proxies, NeurIPS'22 | 69.81 | 51.20 | 95.25 | 96.30 | 92.00 | 91.20 | 72.15 |
| PMRC, CVPR'23 | - | - | 94.80 | 95.40 | 91.80 | - | - |
| **TRD (Ours)** | **70.90** ± 0.22 | **52.15** ± 0.12 | **95.60** ± 0.08 | **96.35** ± 0.03 | **92.10** ± 0.04 | **91.45** ± 0.12 | **72.27** ± 0.05 |

Table 2: Comparison with SOTA on standard small, medium, and large scale FGVC benchmarks.

drops. This happens because of a lack of prior knowledge about the proxy-graph structure, which increases room for noisy views to be more influential, harming classification robustness. To alleviate this, we introduce TR based on our findings from Theorem 3, which effectively wards off the influence of noisy views. With PD, Rows 3 and 4 respectively show the individual contributions of CG, and enforcing the Transitivity invariant between instance and proxy graphs. Row 5 shows that TRD is at its best with all components included.

## 5 Conclusion & Discussion

We presented Transitivity Recovering Decompositions (TRD), an approach for learning interpretable relationships for FGVC. We theoretically showed the existence of semantically equivalent graphs for abstract relationships, and derived their key information theoretic and topological properties. TRD is a search algorithm in the space of graphs that looks for solutions that satisfy the above properties, providing a concrete, human understandable representation of the learned relationships. Our experiments revealed that TRD not only provides end-to-end transparency, all the way up to the class-proxy representation, but also achieves state-of-the-art results on small, medium, and large scale benchmark FGVC datasets, a rare phenomenon in the interpretability literature. We also showed that our method is robust to noisy input views, both theoretically and empirically, which we conjecture to be a crucial factor behind its effectiveness.

**Limitations:** Although robust to noisy views, our method is not fully immune to spurious correlations (Figure 3, rightmost graph – the wing and the sky are spuriously correlated; additional examples in the supplementary). Combined with recent advances in learning decorrelated representations [66], we believe that our method can overcome this limitation while remaining fully interpretable.

**Societal Impacts:** Our method makes a positive societal impact by adding interpretability to existing fine-grained relational representations, in a provably robust manner. Although we are not aware of any specific negative societal impacts that our method might have, like most deep learning algorithms, our method is susceptible to the intrinsic biases of the training set.

## Acknowledgements

This work was supported by the MUR PNRR project FAIR - Future AI Research (PE00000013) funded by the NextGenerationEU.

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

# A  Appendix

Below, we provide some theoretical results and quantification of empirical observations from the existing literature [24, 9, 10] that are used in the main text.

## A.1  Identities of Mutual Information

Let three random variables $\mathbf{x}$, $\mathbf{y}$, and $\mathbf{z}$ form the Markov Chain $\mathbf{x} \to \mathbf{z} \to \mathbf{y}$. Then,

- Positivity:
$$I(\mathbf{x}; \mathbf{y}) \geq 0, I(\mathbf{x}; \mathbf{y}|\mathbf{z}) \geq 0$$

- Chain Rule:
$$I(\mathbf{x}; \mathbf{y}) = I(\mathbf{x}; \mathbf{y}|\mathbf{z}) + I(\mathbf{x}; \mathbf{z})$$

- Data Processing Inequality:
$$I(\mathbf{x}; \mathbf{z}) \geq I(\mathbf{x}; \mathbf{y})$$

## A.2  Information Gap

The Information Gap [10] is the uncertainty that remains about the label information $\mathbf{y}$ given a relation-agnostic representation $\mathbf{z}$. Quantitatively,

$$I(\mathbf{x}; \mathbf{r}|\mathbf{z}) = I(\mathbf{x}; \mathbf{y}) - I(\mathbf{z}; \mathbf{y})$$

An implication of the above is that the label information in $\mathbf{x}$ can be factorized as follows:

$$I(\mathbf{x}; \mathbf{y}) = I(\mathbf{x}; \mathbf{r}|\mathbf{z}) + I(\mathbf{z}; \mathbf{y})$$

## A.3  Sufficiency

A representation $\mathbf{z}$ of $\mathbf{x}$ is sufficient [24] for predicting its label $\mathbf{y}$ if and only if, given $\mathbf{z}$, there remains no uncertainty about the label of $\mathbf{y}$ of $\mathbf{x}$. Formally,

$$I(\mathbf{x}; \mathbf{y}|\mathbf{z}) = 0 \iff I(\mathbf{x}; \mathbf{y}) = I(\mathbf{z}; \mathbf{y})$$

## A.4  Relation-Agnostic Representations - Information Theoretic

An encoder $f$ is said to produce relation-agnostic representations [10] if it independently encodes the global view $\mathbf{g}$ and local views $\mathbf{v} \in \mathbb{L}$ of $\mathbf{x}$ without considering their relationship information $\mathbf{r}$. Quantitatively,

$$I(\mathbf{x}; \mathbf{y}|\mathbf{z}) = I(\mathbf{x}; \mathbf{r}) = I(\mathbf{v}_1\mathbf{v}_2...\mathbf{v}_{|\mathbb{L}|}; \mathbf{g})$$

## A.5  Distributive property of $f$

Since $I(\mathbf{x}; \mathbf{r})$ refers to local-to-global relationships $I(\mathbf{v}_1\mathbf{v}_2...\mathbf{v}_{|\mathbb{L}|}; \mathbf{g})$, the definition of relation agnosticity in Appendix A.4 implies that $\mathbf{z} = f(\mathbf{x})$ does not reduce any uncertainty about emergence, *i.e.*,

$$I(\mathbf{x}; \mathbf{r}) = I(\mathbf{v}_1\mathbf{v}_2...\mathbf{v}_{|\mathbb{L}|}; \mathbf{g}) = I(f(\mathbf{v}_1)f(\mathbf{v}_2)...f(\mathbf{v}_{|\mathbb{L}|}); f(\mathbf{g})) = I(\mathbf{z}_1, \mathbf{z}_2, ...\mathbf{z}_{|\mathbb{L}|}; \mathbf{z}_{\mathbf{g}})$$

## A.6  Sufficient Learner

Following are the necessary and sufficient conditions for a sufficient learner [10]:

- It should have a relation-agnostic encoder $f$ producing mappings $\mathbf{x} \mapsto \mathbf{z}$.

- There must be a separate encoder $\xi$ that takes relation-agnostic representations $\mathbf{z}$ as input, and bridges the information gap $I(\mathbf{x}; \mathbf{r}|\mathbf{z})$.

### A.7 Semantic Consistency

A metric space $(\mathcal{M}, d)$ along with its embedding function $\varphi : \mathbb{X} \to \mathcal{M}$ are called semantically consistent, if the following holds:

$$I(\mathbf{x}; \mathbf{y} | \varphi(\mathbf{x})) \propto d(\varphi(\mathbf{x}), \mathbf{y})$$

In other words, $\mathcal{M}$ encodes the semantics of $\mathbf{x}$ via its distance function $d$.

### A.8 Semantic Emergence

Without loss of generality, let us assume that $\mathbf{g}$ only contains semantic information. Semantic Emergence refers to the fact that if, for a particular view, its embedding $\mathbf{z}_v$ encodes a large amount of semantic information, it must also have a large amount of information about the global-view (local-to-global correspondence in DiNo [9]). Intuitively, this is the case because all views $\mathbf{v} \in \mathbb{V}$, in some way, arise from $\mathbf{g}$, and local-to-global relationships encode semantic information. Quantitatively,

$$d(\mathbf{z_v}; \mathbf{y}) \propto \frac{1}{I(\mathbf{v}; \mathbf{g})}$$

### A.9 Proofs of Theorems

**Theorem 1:** For a relational representation $\mathbf{r}$ that minimizes the information gap $I(\mathbf{x}; \mathbf{y} | \mathbb{Z}_\mathbb{V})$, there exists a metric space $(\mathcal{M}, d)$ that defines a Semantic Relevance Graph such that:

$$d(\mathbf{z}_i, \mathbf{y}) > \delta \wedge d(\mathbf{z}_j, \mathbf{y}) > \delta \Rightarrow$$
$$\exists\, \xi : \mathbb{R}^n \xrightarrow{\text{sem}} \mathbb{R}^n \mid \mathbf{r} = \xi(\varphi(\mathbf{z}_i), \varphi(\mathbf{z}_j)), d(\mathbf{r}, \mathbf{y}) \leq \delta$$

where $\mathcal{M} \in \mathbb{R}^n$, $\xrightarrow{\text{sem}}$ and $d$ denote semantically consistent transformations and distance metrics respectively, $\varphi : \mathbb{Z}_\mathbb{V} \xrightarrow{\text{sem}} \mathcal{M}$, and $\delta$ is some finite, empirical bound on semantic distance.

*Proof.* By contradiction.

Let us assume that there exists no metric space $(\mathcal{M}, d)$ that induces a Semantic Relevance Graph on $\mathbb{Z}_\mathbb{V}$ satisfying the given constraint. In other words, given $\mathbb{Z}_\mathbb{V}$, we get an upper-bound of $\delta$ on the amount of label information $\mathbf{y}$ that can be captured from $\mathbf{x}$, and there is no function $\xi$ that can reduce the uncertainty about the label any further.

Since $f$ and $d$ are semantically consistent (Appendix A.7), in information theoretic terms, $\mathbb{Z}_\mathbb{V}$ is a *sufficient representation* (Appendix A.3), *i.e.*, both sides of the following are true:

$$I(\mathbf{x}, \mathbf{y}) = I(\mathbb{Z}_\mathbb{V}, \mathbf{y}) \propto \delta$$
$$\iff I(\mathbf{x}; \mathbf{y} | \mathbb{Z}_\mathbb{V}) \propto \sum_{\mathbf{z} \in \mathbb{Z}_\mathbb{V}} d(\mathbf{z}, \mathbf{y}) = \delta, \tag{2}$$

where $\delta \geq 0$ (Appendix A.1). However, since $f$ is only a relation-agnostic encoder (as stated in the Preliminaries), it exhibits an information gap (Appendix A.2), *i.e.*,

$$I(\mathbf{x}, \mathbf{y}) = I(\mathbb{Z}_\mathbb{V}, \mathbf{y}) + I(\mathbf{x}; \mathbf{r} | \mathbb{Z}_\mathbb{V}), \tag{3}$$

where $I(\mathbf{x}; \mathbf{r} | \mathbb{Z}_\mathbb{V}) > 0$. This leads to a contradiction between Equation (2) and Equation (3), establishing the falsity of the initial assumption and proving Theorem 1. $\qquad \square$

**Theorem 2:** The graph $\mathcal{G}^*$ underlying the output relational representation is a member of $\mathbb{G}$ for which the mutual information with the label is maximal. Formally,

$$\mathcal{G}^* = \arg\max_{\mathcal{G} \in \mathbb{G}} I(\mathcal{G}, \mathbf{y})$$

*Proof.* By contradiction.

Let us assume the existence of a graph $\mathcal{G}' \in \mathbb{G}$ such that $I(\mathcal{G}', \mathbf{y}) > I(\mathcal{G}^*, \mathbf{y})$. As shown in Theorem 1, since $\mathcal{G}^*$ encodes both relation-agnostic (nodes) and relation-aware (edges) information, it is a sufficient representation (Appendix A.3) of $\mathbf{x}$. Hence, for $\mathbf{z} \in \mathbb{Z}_\mathbb{V}$,

$$I(\mathcal{G}^*; \mathbf{y}) = I(\mathbf{x}; \mathbf{r} | \mathbf{z}) + I(\mathbf{z}; \mathbf{y}) = I(\mathbf{x}; \mathbf{y})$$

Thus, for a $\mathcal{G}'$ to exist,

$$I(\mathcal{G}'; \mathbf{y}) > I(\mathbf{x}; \mathbf{y}) \tag{4}$$

However, computing $\mathcal{G}'$ involves the following Markov Chain:

$$\mathbf{x} \to \mathbb{Z}_{\mathbb{V}} \to \mathcal{G}'$$

Thus, from the Data Processing Inequality (Appendix A.1):

$$I(\mathbf{x}; \mathbb{Z}_{\mathbb{V}}) \geq I(\mathbb{Z}_{\mathbb{V}}; \mathcal{G}')$$

Using the chain rule of mutual information (Appendix A.1):

$$\Longrightarrow I(\mathbf{x}; \mathbf{y}) - I(\mathbf{x}; \mathbf{y}|\mathbb{Z}_{\mathbb{V}}) \geq I(\mathbb{Z}_{\mathbb{V}}; \mathcal{G}')$$
$$\Longrightarrow I(\mathbf{x}; \mathbf{y}) - I(\mathbf{x}; \mathbf{y}|\mathbb{Z}_{\mathbb{V}}) \geq I(\mathcal{G}'; \mathbf{y}) - I(\mathcal{G}'; \mathbf{y}|\mathbb{Z}_{\mathbb{V}})$$

Now, $I(\mathcal{G}'; \mathbf{y}|\mathbb{Z}_{\mathbb{V}}) = 0$, since $\mathbb{Z}_{\mathbb{V}} \to \mathcal{G}'$ ($\mathcal{G}'$ is derived from $\mathbb{Z}_{\mathbb{V}}$),

$$\Longrightarrow I(\mathbf{x}; \mathbf{y}) - I(\mathbf{x}; \mathbf{y}|\mathbb{Z}_{\mathbb{V}}) \geq I(\mathcal{G}'; \mathbf{y}) \Longrightarrow I(\mathbf{x}; \mathbf{y}) \geq I(\mathcal{G}'; \mathbf{y}) + I(\mathbf{x}; \mathbf{y}|\mathbb{Z}_{\mathbb{V}})$$

By the positivity property of mutual information [24], $I(\mathbf{x}; \mathbf{y}|\mathbb{Z}_{\mathbb{V}}) \geq 0$. Hence,

$$I(\mathbf{x}; \mathbf{y}) \geq I(\mathcal{G}'; \mathbf{y}),$$

which is in contradiction to Equation (4). Hence the initial assumption must be false. $\qquad \square$

**Theorem 3:** A function $\varphi(\mathbf{z})$ that reduces the uncertainty $I(\mathbf{v}_i \mathbf{v}_j; \mathbf{g}|\varphi(\mathbf{z}_i)\varphi(\mathbf{z}_j))$ about the emergent relationships among the set of views can define a Semantic Relevance Graph by being transitivity recovering.

*Proof.* Without loss of generality, consider three *transitively* related views $\mathbf{v}_1$, $\mathbf{v}_2$, and $\mathbf{v}_3$ such that:

$$d(\mathbf{z}_2, \mathbf{y}) < \delta \wedge d(\mathbf{z}_1, \mathbf{y}) > \delta \wedge d(\mathbf{z}_3, \mathbf{y}) > \delta$$

In qualitative terms, $\mathbf{v}_2$ plays a strong role in determining $\mathbf{y}$ given $\mathbb{Z}_{\mathbb{V}}$, while the other views do not. From the property of Semantic Emergence (Appendix A.8),

$$d(\mathbf{z}_2, \mathbf{y}) < \delta \Rightarrow I(\mathbf{v}_1\mathbf{v}_2; \mathbf{g}) > \gamma \wedge I(\mathbf{v}_2\mathbf{v}_3; \mathbf{g}) > \gamma$$

However, since $\mathbf{v}_1$, $\mathbf{v}_2$, and $\mathbf{v}_3$ are transitively related,

$$I(\mathbf{v}_1\mathbf{v}_3; \mathbf{g}) > \gamma$$

Now, $f$ being a relation agnostic encoder, it has no way to minimize the uncertainty in the emergence $I(\mathbf{v}_1\mathbf{v}_3; \mathbf{g})$. Hence, using the distributive property of $f$ (Appendix A.5),

$$I(\mathbf{v}_1\mathbf{v}_3; \mathbf{g}) > \gamma \Rightarrow I(\mathbf{z}_1\mathbf{z}_3; \mathbf{z}_\mathbf{g}) > \gamma$$

We know that $I(\mathbf{x}; \mathbf{r}) = I(\mathbf{v}_1\mathbf{v}_3; \mathbf{g}) = I(\mathbf{z}_1\mathbf{z}_3; \mathbf{g})$ (Appendix A.4).

Consider a function $\varphi(\mathbf{z})$ that maps elements in $\mathbb{Z}_{\mathbb{V}}$ to $(\mathcal{M}, d)$ such that $d(\varphi(\mathbf{z}_i), \varphi(\mathbf{z}_j)) \propto 1/I(\mathbf{z}_i\mathbf{z}_j; \mathbf{g})$. $\varphi$ is thus *transitivity recovering*, as it allows for the identification of transitivity among $\mathbf{v}_1$, $\mathbf{v}_2$ and $\mathbf{v}_3$ via the distance metric $d$. By allowing for such a distance computation, $\varphi$ minimizes $I(\mathbf{z}_1\mathbf{z}_3; \mathbf{g}|\varphi(\mathbf{z}_1)\varphi(\mathbf{z}_3))$, meaning that the knowledge obtained from $\varphi$ reduces the uncertainty about the emergent relationships. This in turn also minimizes $I(\mathbf{v}_1\mathbf{v}_3; \mathbf{g}|\varphi(\mathbf{z}_1)\varphi(\mathbf{z}_3))$, and consequently $I(\mathbf{x}; \mathbf{r}|\varphi(\mathbf{z}_1)\varphi(\mathbf{z}_3))$.

However, $I(\mathbf{x}; \mathbf{r}) = I(\mathbf{x}; \mathbf{y}|\mathbf{z}_1\mathbf{z}_3)$ (Appendix A.4). Hence, $\varphi$ also minimizes the information gap $I(\mathbf{x}; \mathbf{y}|\mathbf{z}_1\mathbf{z}_3\varphi(\mathbf{z}_1)\varphi(\mathbf{z}_3))$, which implies that:

$$\exists \xi \mid \mathbf{r} = \xi(\varphi(\mathbf{z}_1), \varphi(\mathbf{z}_3)), d(\mathbf{r}, \mathbf{y}) < \delta,$$

thus defining a Semantic Relevance Graph (Theorem 1). $\qquad \square$

### A.9.1 Robustness

**Definition 4** (**Robustness**). A classifier $f_\eta$ is said to be robust (noise-tolerant) if it has the same classification accuracy as that of $f$ under the distribution $\mathcal{D}$ [26]. Formally,

$$P_\mathcal{D}[f(x)] = P_\mathcal{D}[f_\eta(x_\eta)],$$

where $P_\mathcal{D}$ defines the probability distribution over $\mathcal{D}$, and $x_\eta$ is the noisy version of the input $x$. In other words, $f_\eta$ would base its prediction on the $(1 - \eta)|\mathbb{V}|$ noise-free views only.

**Lemma 1.** *Let $\mathbb{V}^\eta$ be the set of noisy views in $\mathbb{V}$. Let $\mathbb{V}^* = \mathbb{V} - \mathbb{V}^\eta$ be the set of noise free (and hence, transitively related - Theorem 3) views. Then, the cut-set $\mathbb{C} = (\mathbb{V}^*, \mathbb{V}^\eta) = \varnothing$ when mapped to the output space of $\varphi : \mathbb{V} \to \mathbb{G}$.*

*Proof.* $\varphi$ restricts $\mathbb{G}$ to the Hausdorff space (Theorem 3, Inference and Learning objective). In the general case, since the Hausdorff distance defines a pseudometric [54], it satisfies the triangle inequality. In our specific case, it also defines a complete metric. This is because $\varphi$ also satisfies the requirement of geometric relation agnositicity in the fine-grained scenario under the minimization of the cross-entropy loss with a categorical ground-truth [10]:

$$\forall \mathbf{z}_l \in \mathbb{Z}_\mathbb{L} : n_\epsilon(\mathbf{z}_l) \cap n_\epsilon(\mathbf{z}_g) = \phi$$

The above implies that the representation space of $\varphi$ satisfies the separation axioms [57], and hence:

$$h(x, y) > 0, \forall x \neq y,$$

where $h(\cdot, \cdot)$ is the Hausdorff edit distance [25, 64]. Therefore, for $\varphi(\mathbf{z})$, the Hausdorff distance defines a complete metric while satisfying the triangle inequality. Now, consider a set of views $\{\mathbf{v}_i, \mathbf{v}_j, \mathbf{v}_k\}$, where $\mathbf{v}_i, \mathbf{v}_k \in \mathbb{V}^*$, and $\mathbf{v}_j \in \mathbb{V}^\eta$. Suppose these nodes define a 3-clique (clique of size 3) in the instance graph. The completeness of the Hausdorff distance in $\varphi(\mathbf{z})$, as well as the satisfaction of the triangle inequality thus gives us:

$$d(\varphi(\mathbf{v}_i), \varphi(\mathbf{v}_k)) < d(\varphi(\mathbf{v}_i), \varphi(\mathbf{v}_j)) + d(\varphi(\mathbf{v}_j), \varphi(\mathbf{v}_k))$$

Hence, for the aforementioned clique formed by $\mathbf{v}_i, \mathbf{v}_j$, and $\mathbf{v}_k$, the only edge that survives $\varphi$ is $(\mathbf{v}_i, \mathbf{v}_k)$. In other words, upon convergence, $\varphi$ would never produce a path between two transitively related views that involves a noisy view, rendering $\mathbb{V}^*$ and $\mathbb{V}^\eta$ disconnected. This completes the proof of the lemma. □

**Corollary 4.1.** *There is no path in $\mathcal{G}^*$ connecting two views in $\mathbb{V}^*$ that involves a noisy view from $\mathbb{V}^\eta$. Thus, the elements of $\mathbb{V}^*$ and $\mathbb{V}^\eta$ respectively form homogeneous subgraphs of transitive and noisy views in $\mathcal{G}^*$ and $\mathcal{G}_\mathbf{p}$.*

**Definition 5** (**Continents and Islands**). A *continent* is a complete subgraph (clique) in $\mathcal{G}^*$ exclusively composed of transitively related views. An *island* in $\mathcal{G}^*$ is a subgraph that exclusively contains noisy views.

**Lemma 2.** *Let $\mathbb{E}_{12}$ be the set of edges connecting continents $\mathbb{C}_1$ and $\mathbb{C}_2$. The elements of $\mathbb{E}_{12}$ are mutually redundant, i.e.,*

$$H(e_j|e_i) = 0, \ \ \forall(e_i, e_j) \in \mathbb{E}_{12}$$

*Proof.* Consider a pair of vertices $v_i, v_j \in \mathbb{C}_1$, such that both are connected to $\mathbb{C}_2$ via edges $e_i$, and $e_j$ respectively. However, since $\mathbb{C}_1$ is a clique, $v_i$ and $v_j$ (and all other nodes in $\mathbb{C}_1$) are also connected. Thus, the complete set of information that $\mathbf{v}_i$ has access to (and can potentially encode via $e_i$) is the transitivity of (local-to-global relationship emerging from) all the local views in $\mathbb{C}_1$ given by:

$$H(e_i) = I(\mathbf{v_1}, \mathbf{v_2}, ..., \mathbf{v}_{|\mathbb{C}_1|}; \mathbf{g})$$

Let $\gamma$ be a message passing function (like a GNN) that carries information across connected nodes. Thus, for $\gamma$, the following holds:

$$e_j \equiv v_j \xrightarrow{\gamma} v_i \xrightarrow{\gamma} e_i \implies I(e_j; \mathbb{C}_2) = I(e_i; \mathbb{C}_2) \implies H(e_j|e_i) = 0,$$

meaning that the knowledge of a single edge between $\mathbb{C}_1$ and $\mathbb{C}_2$ is sufficient to determine the connectivity of the two continents. No other edge in $\mathbb{E}_{12}$ can thus provide any additional information, thereby concluding the proof. □

**Theorem 4:** In the representation space of $\varphi$, the uncertainty in estimating the topology $\tau^\eta$ of $\mathbb{V}^\eta$ is exponentially greater than the uncertainty in estimating the topology $\tau^*$ of $\mathbb{V}^*$, in terms of the error rate $\epsilon$.

*Proof.* It is known that the total number of possible topologies $\tau$ for a graph with $n$ nodes is given by [40]:

$$|\tau| = 2^{(n^2/4)} \tag{5}$$

In what follows, we show that it is possible to dramatically reduce this number to $\mathcal{O}(n)$ for continents, but not for islands.

According to Lemma 1 and Corollary 4.1, the groups of transitive and noisy view nodes can be separated into disjoint sets with no edges between them. Let $\mathbb{V}^* = \{\mathbf{v}_1, \mathbf{v}_2, ..., \mathbf{v}_k\}$ be a set of transitively related views (a continent) with topology $\mathcal{T}$. Let $\mathbb{V}^\eta = \{\mathbf{v}_{k+1}, \mathbf{v}_{k+2}, ..., \mathbf{v}_n\}$ be a set of noisy views (an island) with topology $\mathcal{T}_\eta$. Let $\pi^\tau$ denote the probability mass of a topology $\tau$ in $\mathcal{D}$ (fraction of samples in $\mathbb{X}$ with topology $\tau$).

Let $\tau^*$ be the ground truth topology of a class, determined exclusively by the continents (since noisy views should not determine the label). Let $\hat{\tau}$ be the best approximation of $\tau^*$ that can be achieved by observing $M$ *unique* samples, such that:

$$h[\mathcal{G}(\tau^*) - \mathcal{G}(\hat{\tau})] \leq \epsilon,$$

where $\mathcal{G}(\tau)$ is the graph with topology $\tau$. In other words, $\epsilon$ is the upper-bound on the statistical error in the approximation of the ground truth topology that can be achieved with $M$ unique samples.

Now, uniquely determining $\mathcal{G}(\tau^*)$ involves determining all the $k$-cliques involving the global-view (Transitivity Recovery from Theorem 3). However, the Túran's theorem [74] puts an upper-bound on the number of edges $|\mathbb{E}|$ of a graph with clique number $k$ and $n$ nodes at:

$$|\mathbb{E}| = (1 - \frac{1}{k})\frac{n^2}{2},$$

which gives a direct upper-bound on the sample complexity $M$ for learning $\hat{\tau}$.

If there are $|\mathbb{E}|$ edges representing transitive relationships among $n$ views, and the maximum clique size for a class is $k$, the number of $k$-cliques to encode all possible clique memberships with this configuration would be given by:

$$\log_k e = \frac{\log_2 e}{\log_2 k} = \frac{\log_2[(1 - \frac{1}{k})\frac{n^2}{2}]}{\log_2 k} = \frac{\log_2[(1 - \frac{1}{k})n^2]}{\log_2 k} = \log_2(1 - \frac{1}{k}) - \log_2 k + 2\log_2 n = \mathcal{O}(\log_2 n) \tag{6}$$

Since the maximum clique size $k$ is a constant for a given class. The above number of samples would be required to gain the knowledge of all the cliques in the graph.

To analyze the connectivity across different continents in $G^*$, we can apply Lemma 2 to reduce all the continents in $G^*$ into single nodes, connected by an edge if there is at least one node connecting the two continents in $G^*$. Using this observation, and Lemma 2, the number of topologies $|\mathcal{R}|$ of such a graph can be given by:

$$|\mathcal{R}| = 2^{(\mathcal{O}(\log_2 n)/4)} = \mathcal{O}(n) \tag{7}$$

Thus with a clique number of $k$, $\mathcal{G}^*$ could have a maximum of $\mathcal{O}(\log_2 n)$ continents (Equation (6)), connected with each other in $\mathcal{O}(n)$ possible ways (Equation (7)). Since there is only 1 possibility for the topologies of each clique, the only variation would come from the cross-continent connectivity, which has $\mathcal{O}(n)$ potential options. Using the Haussler's theorem [28], the number of samples $M$ required to achieve a maximum error rate of $\epsilon$ with probability $(1 - \delta)$ for a continent is given by:

$$M^* = \frac{\log_2 \mathcal{O}(n) + \log_2 \frac{1}{\delta}}{\epsilon \pi^{\hat{\tau}}} = \frac{\log_2 \mathcal{O}(n) + \log_2 \frac{1}{\delta}}{\epsilon(1/c)} \approx \frac{\log_2 \mathcal{O}(n) + \log_2 \frac{1}{\delta}}{\epsilon}, \tag{8}$$

where $c$ is the number of classes, and in practice $1/c \gg 0$. Note that the above bound is tight since we restrict our hypothesis class to only having transitivity recovering decompositions (Theorem 3).

Thus, one would need a number of samples that is linear in the number of transitive views ($n$), as well as the the error rate ($\epsilon$), and the certainty ($\delta$), to determine $\hat{\tau}$ for a class. It can also be seen that $M^*$ is completely independent of the noise rate $\eta$.

However, for $\mathbb{V}^\eta$, since its constituent views are noisy, no prior assumptions can be made about its structure. Thus, the number of possible topologies for $\mathbb{V}^\eta$ can be obtained by substituting $n$ with $\eta$ in Equation (5) as follows:

$$|\tau^\eta| = 2^{(\eta^2/4)} = \mathcal{O}(2^\eta)$$

Thus, based on the Haussler's theorem, the sample complexity for learning the topology of an island would be given by:

$$M^\eta \geq \frac{\log_2 \mathcal{O}(2^\eta) + \log_2 \frac{1}{\delta}}{\epsilon \pi^{\tau^\eta}} \approx \frac{\log_2 \mathcal{O}(2^\eta) + \log_2 \frac{1}{\delta}}{\epsilon \cdot \epsilon} = \frac{\log_2 \mathcal{O}(2^\eta) + \log_2 \frac{1}{\delta}}{\epsilon^2}, \qquad (9)$$

since the probability of finding an instance with a specific noisy topology $\pi^{\tau^\eta}$ is very small, and hence can be approximated with the error-rate ($\epsilon$). Note that the small probability mass for a specific $\tau^\eta$ comes from the worst-case assumption that the dataset would have all possible forms of noise, and hence, the probability for a specific kind would be very low. Also note that unlike Equation (8), the bound cannot be tightened as the premise of transitivity does not hold.

Thus, while the sample complexity for learning a transitive topology $M^*$ is linear in the number of views and the error rate Equation (8), that for a noisy topology $M^\eta$ is exponential in the number of views and quadratic in the error rate Equation (9). Hence, in the worst case, when $n \approx \eta$ [26], $M^\eta \gg M^*$. This implies that it is significantly more difficult for $\varphi$ to learn the topology $\tau^\eta$ of an island than it is to learn the topology $\hat{\tau}$ of a continent.

From Equation (9), the probability of $\varphi$ making a correct prediction based on noisy views would be given by:

$$M_\eta \geq \frac{\log_2 \mathcal{O}(2^\eta) + \log_2 \frac{1}{\delta}}{\epsilon^2} \implies \log_2 \frac{1}{\delta} \leq M_\eta \epsilon^2 - \log_2 \mathcal{O}(2^\eta)$$

$$\implies \log_2 \frac{1}{\delta} \leq M_\eta \epsilon^2 - \eta$$

$$\implies \frac{1}{\eta} \log_2 \frac{1}{\delta} \leq \frac{M_\eta}{\eta} \epsilon^2 - 1 \approx \epsilon^3 - 1$$

$$\implies \log_2 \frac{1}{\delta} \leq \eta \epsilon^3 - \eta$$

$$\implies \frac{1}{\delta} \leq 2^{\eta(\epsilon^3 - 1)}$$

$$\implies \delta_\eta \geq \mathcal{O}(2^{-\epsilon^3})$$

where the noise rate $\eta$ is a constant for a particular dataset, and $M_\eta/\eta \approx \epsilon$, since $M_\eta \gg \eta$, as $M_\eta = \mathcal{O}(2^\eta)$. From Equation (8), the probability of $\varphi$ making a correct prediction based on transitive views would be given by:

$$M_* = \frac{\log_2 \mathcal{O}(n) + \log_2 \frac{1}{\delta}}{\epsilon} \implies \log_2 \frac{1}{\delta} = M_* \epsilon - \log_2 \mathcal{O}(n) \approx M_* \epsilon - \log_2 n$$

$$\implies \log_2 \frac{1}{\delta} = (\theta \log_2 n)\epsilon - \log_2 n$$

$$\implies \log_2 \frac{1}{\delta} = \log_2 n^{\theta \epsilon} - \log_2 n$$

$$\implies \log_2 \frac{1}{\delta} = \log_2 \frac{n^{\theta \epsilon}}{n} = \log_2 n^{\theta \epsilon - 1}$$

$$\implies \frac{1}{\delta} = n^{\theta \epsilon - 1} - 2^{\Gamma \epsilon - 1}$$

$$\implies \delta_* = \mathcal{O}(2^{-\epsilon}),$$

where $\theta = M_*/\log_2 n$, which is a constant for a particular class in a dataset, and so is the number of transitively related views $n = |\mathbb{V}^*| = 2^l$ for a class, and $\Gamma = \theta l$. Thus the relative uncertainty

between $\delta_\eta$ and $\delta_*$ would be given by:

$$\frac{\delta_\eta}{\delta_*} \geq \frac{\mathcal{O}(2^{-\epsilon^3})}{\mathcal{O}(2^{-\epsilon})} \approx \frac{2^{-\epsilon^3}}{2^{-\epsilon}} = \frac{(2^{-\epsilon})^{\epsilon^2}}{2^{-\epsilon}} = (2^{-\epsilon})^{\epsilon^2 - 1} = 2^{-\epsilon^3 + \epsilon} = \mathcal{O}(2^{-\epsilon^3})$$

This completes the proof of the theorem. $\qquad\square$

## A.10 Discussion on Graph-Based Image Representations and Proxy-Based Graph Metric Learning

Graph-based image representation using patches was recently proposed in ViG [27]. Although the initial decomposition of an image into a graph allowed for flexible cross-view relationship learning, the image graph is eventually abstracted into a single representation in $\mathbb{R}^n$. This prevents one from identifying the most informative subgraph responsible for prediction, at both the instance level, and at the class-level. Although interpreting ViGs may be possible via existing GNN explainablity algorithms [88, 51, 82], they are typically post-hoc in nature, meaning that only the output predictions can be explained and not the entire classification pipeline. Incurred computational and time overheads that come with post-hoc approaches in general, are additional downsides. Our approach preserves the transparency of the entire classification pipeline by maintaining a graph-based representation all the way from the input to the class-proxy, thereby allowing us to explain predictions at both instance and class levels in an ante-hoc manner.

Although graph metric learning has been shown to be effective for learning fine-grained discriminative features from graphs [45], instance-based contrastive learning happens to be (1) computationally expensive and (2) unable to capture the large intra-class variations in FGVC problems [56, 39]. Proxy-based graph metric learning [99] has been shown to address both issues, while learning generalized representations for each class. However, the metric space embeddings learned by the above methods are not fully explainable in terms of identifying the most influential subgraph towards the final prediction, which is an issue we aim to address in our work.

## A.11 Role of Transitivity

The idea of transitivity allows us to narrow down sets of views whose relation-agnostic information have been exhaustively encoded in $\mathbb{Z}$, and the only available sources of information comes only when all the views are observed *collectively* as the emergent relationship. On the other hand, dealing with a general local-to-global relationship recovery would additionally include view-pairs, *only one* of which might be contributing to the relational information ($I(\mathbf{v}_1\mathbf{v}_2; \mathbf{g})$ and $I(\mathbf{v}_2\mathbf{v}_3; \mathbf{g})$), thus not exhibiting emergence. Using only the former in our proof helps us to identify the set of nodes responsible for bridging the information gap (via emergence) as the ones satisfying transitivity.

## A.12 Learning the Graph Edit Costs

The GNN proposed in [64] would directly learn the mapping $\phi : \mathbb{Z}_\mathbb{V} \to \mathbb{R}$ for computing the edit costs. In TRD, we instead split the GNN for edit cost computation into a **prefix network** $\varphi : \mathbb{Z}_\mathbb{V} \to \mathcal{M}$ and a **cost head** $\psi : \mathcal{M} \to \mathbb{R}$. Note that this factorization is just a different view of $\phi$ that allows us to draw parallels of our implementation with the theory of TRD.

## A.13 Further comparisons with Relational Proxies

**Same number of local views:** We evaluate both Relational Proxies and TRD with the same number of local views in the normal (no explicit addition of noise) setting on the FGVC Aircraft dataset, and report our findings in Table 4. TRD marginally outperforms Relational Proxies for all values of the number of local views, and exhibits a trend of scaling in accuracy with increasing number of local views. Relational Proxies, on the other hand, does not seem to benefit from increasing the number of local views, possibly due to its lack of robustness to noisy views.

**Compute cost:** In Table 5, we provide the computational costs of Relational Proxies and TRD in terms of wall clock time evaluated on FGVC Aircraft (same experimental settings including the number of local views). We can see that TRD is significantly more efficient than Relational Proxies in terms of both single sample inference as well as training time until convergence. This is because of the following reasons:

| # Local Views | 8 | 16 | 32 | 64 |
|---|---|---|---|---|
| Relational Proxies | 95.25 | 95.30 | 95.29 | 95.31 |
| **TRD (Ours)** | **95.27** | **95.45** | **95.52** | **95.60** |

Table 4: Comparison of TRD with Relational Proxies under the same number of local views.

- The Complementarity Graph in TRD is constructed exactly once before training, and the semantic relevance graph, as well as the proxy graph are learned as part of the training process.
- TRD does not involve updating the relation-agnostic encoder, which is a ResNet50, as part of the training process. Relational Proxies requires it to be updated, thereby exhibiting computationally heavier forward (as local view embeddings cannot be pre-computed) and backward passes.

|  | Average Inference Time (ms) | Training Time (hrs) |
|---|---|---|
| Relational Proxies | 130 | 22 |
| **TRD (Ours)** | 110 | 15 |

Table 5: Compute cost of TRD relative to Relational Proxies.

## A.14    Over-smoothing

To understand whether the process of modelling emergent relationships is vulnerable to the over-smoothing phenomenon in GNNs, we evaluated TRD using GATs of up to 64 layers on FGVC Aircraft, presenting our findings in Table 6. We see that the performance does drop as the number of layers are increased beyond 8. To validate whether this is due to the oversmoothing phenomenon, we measure the degree of distinguishability among the nodes by taking the average of their pairwise $L^2$ distances. The table shows that the distinguishability also decreases as we increase the number of layers, suggesting that the over-smoothing phenomenon does occur. Under the light of the above experiments, the 8-layer GAT is an optimal choice for our problem.

| GAT-Depth | 4 | 8 | 16 | 32 | 64 |
|---|---|---|---|---|---|
| Accuracy | 95.05 | **95.60** | 95.32 | 94.78 | 94.20 |
| Distinguishability | 0.87 | 0.63 | 0.49 | 0.21 | 0.09 |

Table 6: Effect of over-smoothing in GAT on learning emergent relationships.

## A.15    Results on ImageNet subsets

Following existing FGVC literature [10, 21], we evaluate the contribution of our novel Transitivity Recovery objective in the coarse-grained (multiple fine-grained subcategories in a single class) and fine-grained subsets of ImageNet, namely Tiny ImageNet [44] and Dogs ImageNet (Stanford Dogs [38]) respectively, and report our findings in Table 7. Although our method can surpass existing SOTA in both the settings, larger gains ($\Delta$) are achieved in the fine-grained setting, suggesting that TRD is particularly well suited for that purpose.

## A.16    Dependence on input image size

We follow recent SOTA FGVC approaches that use relation-agnostic encoders to extract global and local views [10, 94, 84]. In particular, our view extraction process is exactly the same as Relational Proxies [10], with the same input image resolution and backbone (relation-agnostic) encoder. The

|  | Tiny ImageNet | Dogs ImageNet |
|---|---|---|
| **MaxEnt** [21] | 82.29 | 75.66 |
| **Relational Proxies** [10] | 88.91 | 92.75 |
| $a$ : **w/o Transitivity Recovery** | 88.10 | 91.03 |
| $b$ : **with Transitivity Recovery** | 89.02 | 93.10 |
| $\Delta = (b - a)$ | 0.92 | **2.07** |

Table 7: Coarse-grained vs fine-grained classification results.

above approaches first extract the global view from the input image, and crop out local views from the global view. Since there are two scales at which the image is cropped, all the crops are resized to 224x224. In practice, this provides a similar resolution to resizing the full input image to a higher resolution at the start. To evaluate this hypothesis, we re-trained and re-evaluated our model with the input images resized to 448x448, keeping the remainder of the process of view extraction the same. We provide our results on multiple datasets in Table 8, which shows that the performance gains at the higher input resolution are minor, thus supporting our hypothesis.

|  | Soy | FGVC Aircraft | Stanford Cars |
|---|---|---|---|
| **TRD: $224 \times 224$** | 52.15 | 95.60 | 96.35 |
| **TRD: $448 \times 448$** | 52.23 | 95.62 | 96.39 |

Table 8: Dependence of TRD on input image size.

## A.17 Results with Causal Interventions

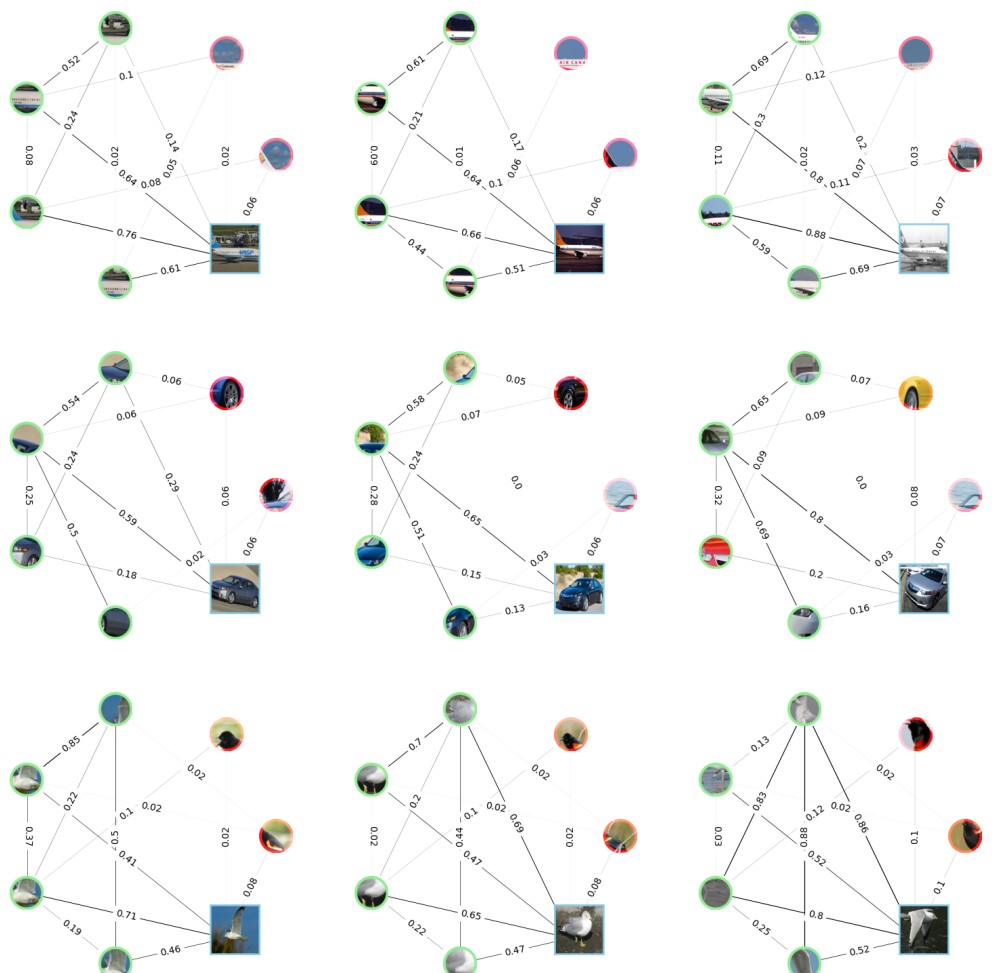

Figure 6: Top-down: Qualitative results on FGVCAircraft, Stanford Cars, and NA Birds with causal intervention. The first two columns from the left are instance graphs, and the rightmost column contains the proxy graphs. The local views of the correct class are bordered in green, while the ones from a different class are bordered in red. A subset of the local views for each instance was corrupted with the introduction of local views from other classes, both during training and inference. These local views consequently also feature in the proxy graphs. The negative local views from different classes (red) can be seen to have a significantly weaker connection to the global view (bordered in blue), and the correct set of local views (green).

