# 1    Qualitative Results from Transitivity Recovering Decompositions (TRD)

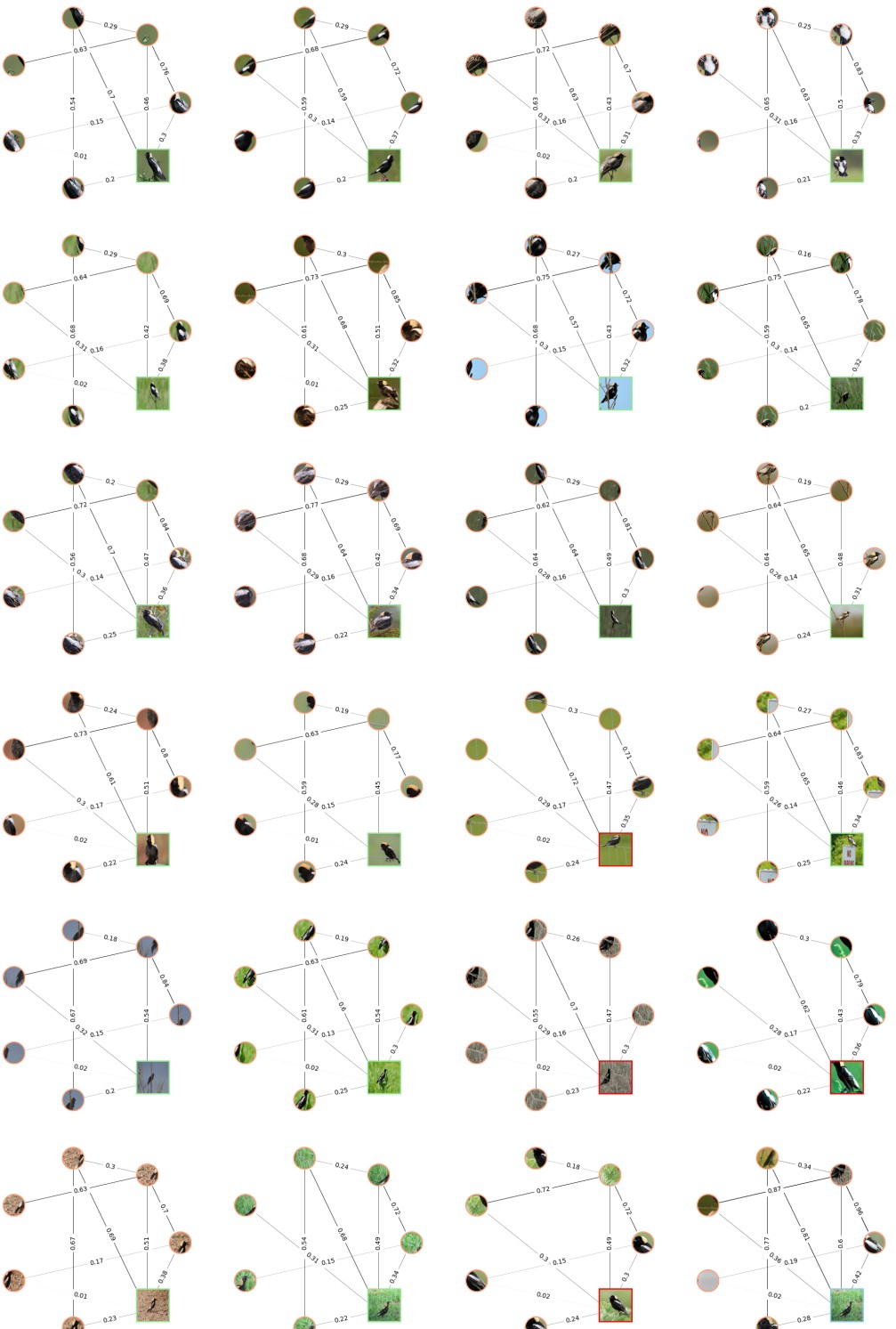

Figure 1: Sample relational interpretation graphs from TRD on CUB. The global views of the correct classifications are bordered with green, and the incorrect ones with red. The global view of the class-proxy graph is colored in blue.

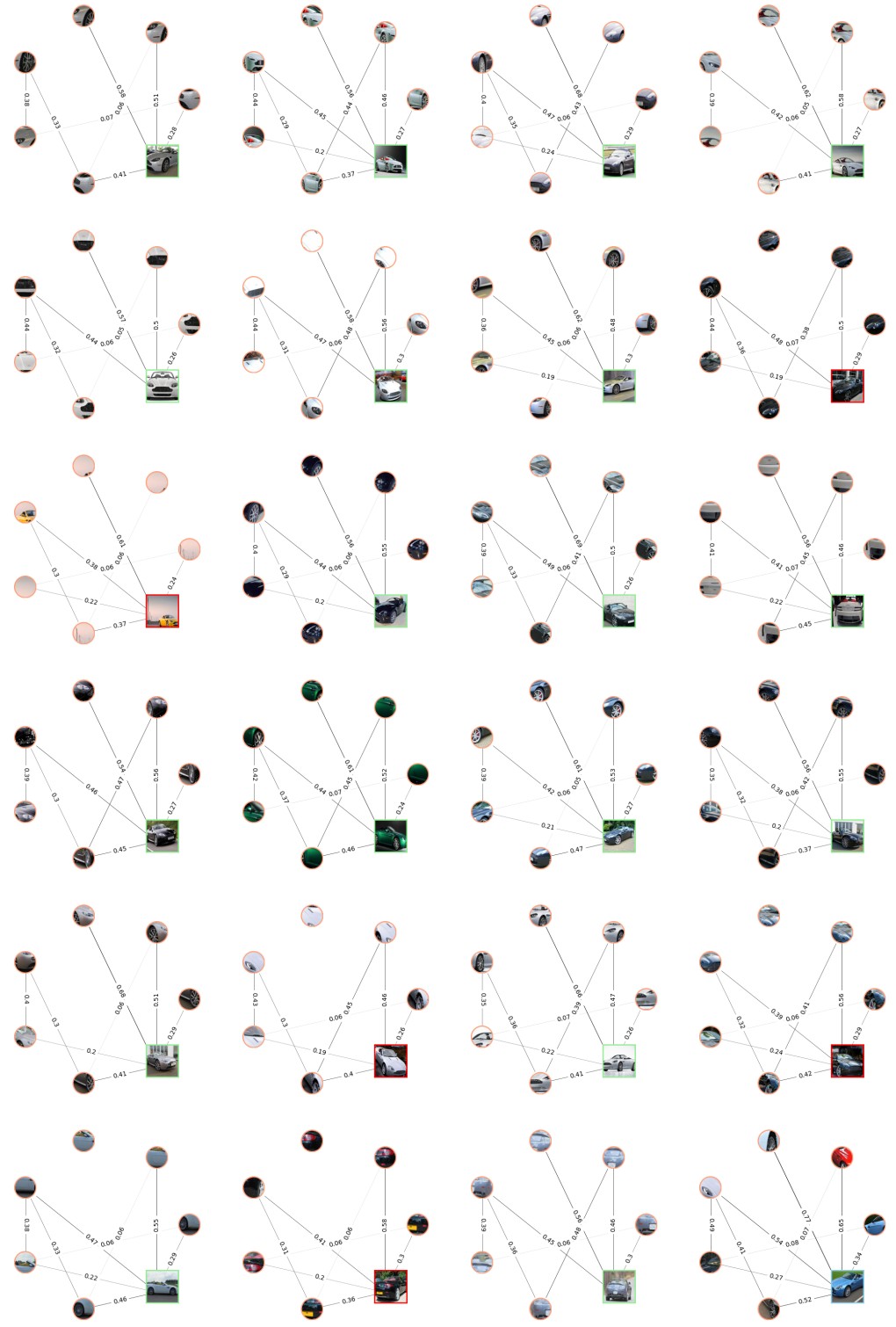

Figure 2: Sample relational interpretation graphs from TRD on Stanford Cars. The global views of the correct classifications are bordered with green, and the incorrect ones with red. The global view of the class-proxy graph is colored in blue.

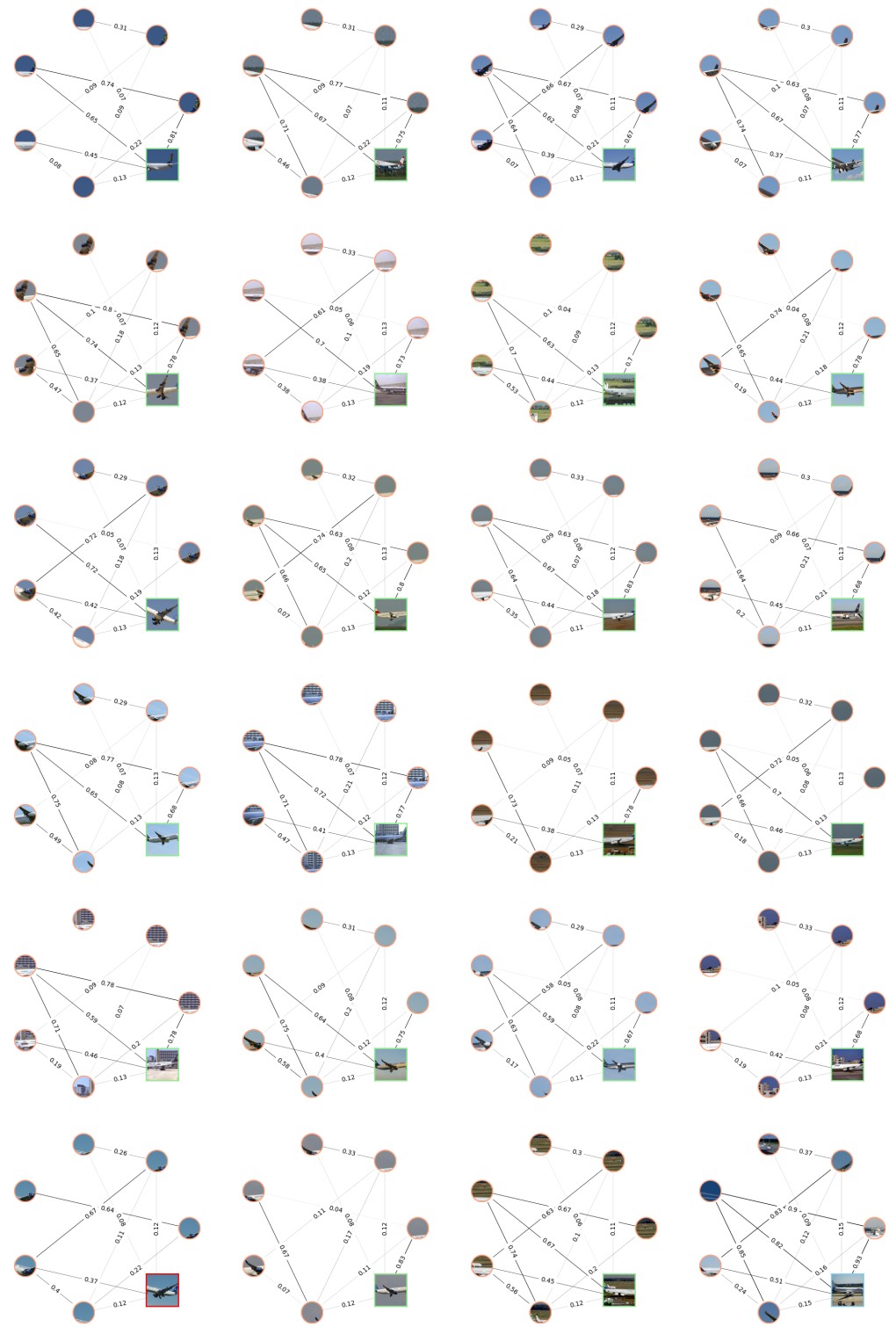

Figure 3: Sample relational interpretation graphs from TRD on FGVC Aircraft. The global views of the correct classifications are bordered with green, and the incorrect ones with red. The global view of the class-proxy graph is colored in blue.

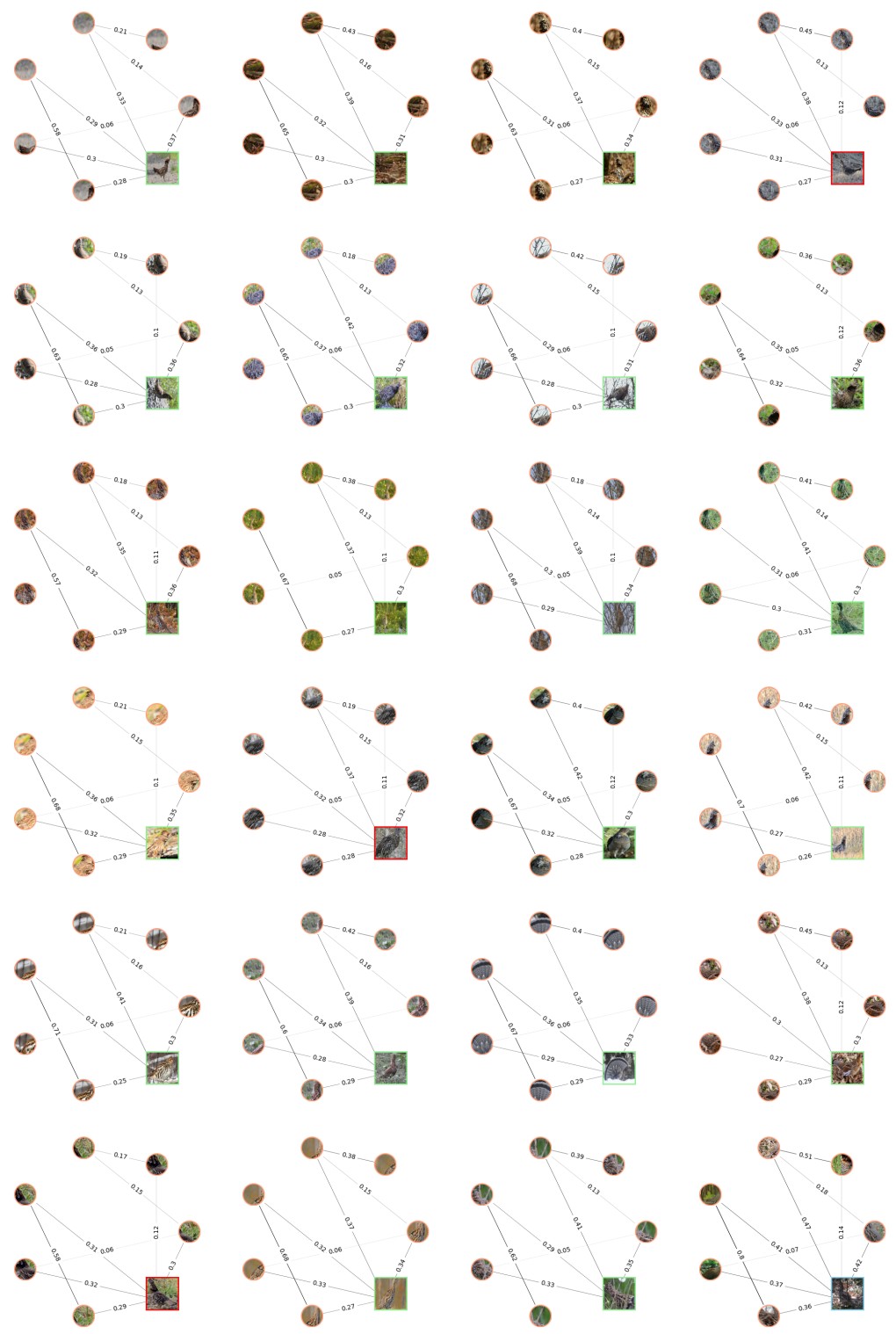

Figure 4: Sample relational interpretation graphs from TRD on NA Birds. The global views of the correct classifications are bordered with green, and the incorrect ones with red. The global view of the class-proxy graph is colored in blue.

## 2  Fidelity *vs* Sparsity Experiments

Figure 5 shows the Fidelity *vs* Sparsity curves on the remaining datasets, *i.e.*, Soy Cultivar, CUB, Stanford Cars, and NA Birds. It can be seen that TRD gives consistently better fidelity over generic GNN explainers across all sparsity levels, and across all datasets, while being fairly stable.

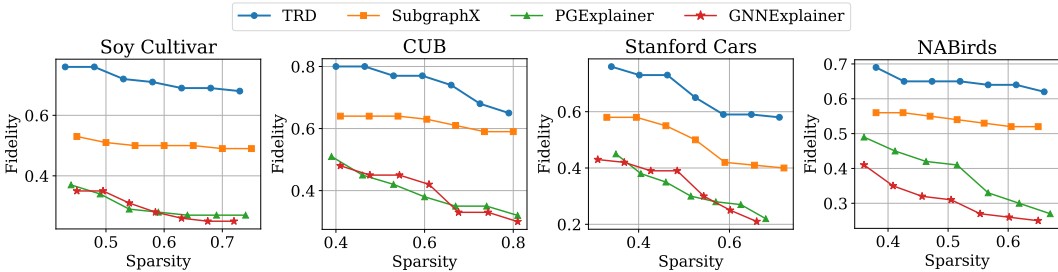

Figure 5: Fidelity *vs* Sparsity curves of TRD and SOTA GNN interpretability methods on datasets apart from the ones presented in the main manuscript.

We follow [6] and use [3] for obtaining the fidelity and sparsity scores for all the methods. Specifically, the fidelity and sparsity values are given by:

$$\texttt{fidelity} = \frac{1}{N} \sum_N h(\mathcal{G}'_s, \mathcal{G}_p) - h(\mathcal{G}_s, \mathcal{G}_p)$$

$$\texttt{sparsity} = \frac{1}{N} \sum_N 1 - \frac{|\mathcal{G}'_s|}{\mathcal{G}_s}$$

where $\mathcal{G}'_s$ is the subgraph of the semantic relevance graph $G_s$ with highest emergence, and $\mathcal{G}_p$ is the proxy graph, and N is the number of test set samples. Also, as noted in [6] as well, since the sparsity values cannot be fully controlled, we obtain the plots in approximately similar sparsity ranges for all methods.

## 3  Equivalence of TRD to Post-Hoc Explanations

**Obtaining Post-Hoc Relational Explanations:** We assume access to the relational embeddings $\mathbf{r}$, and the class-proxy embeddings $\mathbb{P}$ of the source model [2, 1] for which we are trying to generate post-hoc explanations. We start by obtaining a local-view graph $\mathcal{G}_L$ as per [2], which we propagate through a GNN encoder $\varphi' : \mathbb{Z}_\mathbb{V} \to \mathcal{M}$. We partition the nodes of $\varphi'(\mathcal{G}_L)$ into positive and negative sets based on their corresponding similarities with the predicted class-proxy embedding $\mathbf{p} \in \mathbb{P}$ obtained from the source model. Based on this partition, we train $\varphi'$ to contrastively align the positive views with the relational embedding $\mathbf{r}$ of that instance obtained from the source model, while pushing apart the negative ones.

To ensure that the ante-hoc nature of TRD does not affect its explainability accuracy, we compare its performance with post-hoc explanations. To additionally ensure the generalizability of our model, we perform such comparisons with post-hoc explanations obtained across multiple existing abstract relational learning models – Relational Proxies [2] and DiNo [1], that provide SoTA performance in learning fine-grained visual features. Due to the unique nature of our problem of capturing the number of transitively emergent subgraphs in an explanation, the usual metric of AUC [5, 4] for quantitatively evaluating GNN explanations is too simplistic for our setting. For this purpose, to quantify the equivalence of relational explanations obtained from two different algorithms, we propose a metric based on counting the number of $k$-cliques in the explanation graphs, which we call mean Average Clique Similarity (mACS) @ $k$.

For a particular value of $k$, we count the number of $k$-cliques *containing the global view* in the explanation graphs for a particular class and average them, obtaining the Average Clique Count (ACC) for that class. We do this individually for the two algorithms. We take the absolute value

of the difference of the ACCs obtained from the two algorithms, divide it by the greater of the two ACCs, and call this the Average Clique Difference (ACD). We then compute the mean of the ACDs across all classes and subtract it from 1 to get the mACS for the two algorithms. We only consider cliques containing the global view because the generated explanations are designed so as to explicitly capture the learned local-to-global relationships.

We vary the value of $k$ and analyze the mACSs between TRD and the post-hoc explanations for the aforementioned algorithms. The results are reported in Table 1. It can be seen that the explanations obtained from TRD are highly similar to the post-hoc explanations for the SoTA abstract relational embeddings. The mACSs between TRD and DiNo are slightly lower because DiNo does not model the relation-agnostic and relation-aware representation spaces independently as is required for a sufficient learner (Appendix A.6).

| mACS @ | $k = 4$ | $k = 8$ | $k = 12$ | $k = 16$ |
|---|---|---|---|---|
| **Relational Proxies** | 0.99 | 0.99 | 0.96 | 0.95 |
| **DiNo** | 0.87 | 0.85 | 0.85 | 0.81 |

Table 1: Post-hoc equivalence of TRD to existing methods with increasing clique number $k$.