# OpenReview forum: "Transitivity Recovering Decompositions: Interpretable and Robust Fine-Grained Relationships"
_NeurIPS.cc/2023/Conference — NeurIPS 2023 poster_

### Official Review · Reviewer_VJoJ · 2023-07-06

**Soundness:** 3 good
**Presentation:** 3 good
**Contribution:** 3 good
**Rating:** 6
**Confidence:** 3

**Summary:**

This paper aims at fine-grained representation learning. The authors state that local-to-global relationships leveraged in recent fine-grained visual categorization (FGVC) works are abstract. To make such abstract relational representations more human-understandable, the authors first theoretically show the existence of semantically equivalent graphs for abstract relationships and derive their key information theoretic and topological properties. Then, the authors present Transitivity Recovering Decompositions (TRD), which is a graph-space search algorithm that identifies interpretable equivalents of abstract emergent relationships at both instance and class levels and with no post-hoc computations. The authors run experiments to demonstrate the effectiveness of their methods.

**Strengths:**

+ This paper is well-written and easy to follow.
+ The motivation is strong, and the technique in this paper is solid.
+ The proposed method reaches SOTA performance on standard small, medium, and large scale FGVC benchmarks. The authors conduct ablation studies and present visualization results to show the effectiveness of their method.

**Weaknesses:**

- The authors state that the local-to-global (emergent) relationships leveraged in existing methods are abstract fashion. However, no detailed and deep explanation for the term "abstract" is provided in this paper.
- The authors propose a graph-based model for fine-grained visual categorization (FGVC) recognition. However, the authors do not provide a comprehensive review of the relevant literature in the field, such as [1-3], and do not compare their approach with other relevant methods in their experiments.
- The author's comparison seems unfair. In the field of FGVC, a common practice is to resize images to 448$\times$448, but the authors do not follow this approach. Furthermore, there is a lack of experimental details, which raises concerns about the validity of the experiments. I am unsure whether the pre-trained models used in the study are sourced from ImageNet-1K or ImageNet-21K. Additionally, I find the results of TransFG and FFVT on the Aircraft dataset in Table 1 confusing. While I understand that ViT-based models may have limitations on the current dataset, the reported results seem unexpected.

[1] Where to Focus: Investigating Hierarchical Attention Relationship for Fine-Grained Visual Classification, ECCV22\
[2] Weakly Supervised Posture Mining for Fine-Grained Classification, CVPR23\
[3] SR-GNN: Spatial Relation Aware Graph Neural Network for Fine-Grained Image Categorization, TIP

**Questions:**

Please see weaknesses

**Limitations:**

The authors mentioned the limitations and societal impacts of their work.

---

> ### Author Rebuttal · Authors · 2023-08-05
>
> **1. Definition of abstract** Although we briefly refer to what we mean by “abstract” in  Line 23-24, we agree that it requires further elaboration and detailing. We provide an in-depth definition below, which we will also add to the final version.
> Existing works that leverage relational information for representation learning typically adopt the following approaches:
> - Modelling all possible ways the local views can combine to form the global view through a transformer, and distill out the information about the most optimal combination in the summary embedding [8, 9].
> - Modelling the underlying relations through a GNN, but performing an aggregation on its outputs [91, 25]. \
> Both 1 and 2 produce vector valued outputs, and such, cannot be decoded in a straightforward way to get an understanding of what the underlying emergent relationships between views are. This lack of transparency is what we refer to as “abstract”. The above set of methods exhibit this abstraction not only at the instance, but at the class-level as well, which also appear as vector-valued embeddings. On the contrary, an interpretable relationship encoder should be able to produce graphs encoding relationships in the input, intermediate and output spaces, while also representing a class as relationships among concepts, instead of single vectors that summarize information about emergence.
>
> **2. Related works** We thank the reviewer for pointing us to the related works [a, b, c], and we apologize for missing these out in our literature survey. The table below shows a performance comparison on the common benchmarks between ours and [a, b, c]. Due to the short rebuttal timeframe, we were unable to evaluate all of these methods on the remainder of our datasets. However, on all of the common datasets, our TRD can still be seen to surpass all of [a, b, c], thanks to its inherent robustness. **Note that we designed TRD not with the objective of classification accuracy improvement, but rather for providing interpretability to existing relational representation based algorithms.** \
> [a] is a discriminative part discovery based approach and does not involve computing any cross-view relationships. Although both [b] and [c] provide graph-based intermediate image representations, they aggregate the full graph into a single vector-valued output, keeping the final image and class representations abstract.\
> We will add the above discussions on [a, b, c] in section 4.3 as well as the below comparison to Table 1 in the final version.
>
> |                  |    CUB    | FGVC Aircraft | Stanford Cars |
> |------------------|:---------:|:-------------:|:-------------:|
> | WhereToFocus [a] |   90.80   |     94.70     |     95.30     |
> | PMRC [b]         |   91.80   |     94.80     |     95.40     |
> | SR-GNN [c]       |   91.90   |     95.40     |     96.10     |
> | **TRD (Ours)**   | **92.10** |   **95.60**   |   **96.35**   |
>
> **3. Input size, pretrained weights, and performance of transformers**
> - Input size - It is true that certain works on FGVC that consider the complete image as the only input to the model do resize the image to 448x448. However, we follow recent SoTA FGVC approaches that use relation-agnostic encoders to extract global and local views [9, 78, 88]. In particular, our view extraction process is exactly the same as Relational Proxies (NeurIPS 2022) [9], with the same input image resolution and backbone (relation-agnostic) encoder. The above approaches first extract the global view from the input image, and crop out local views from that. Since there are two scales at which the image is cropped, all the crops are resized to 224x224. In practice, this provides a similar resolution to resizing the full image to 448x448.
> Following the reviewer’s proposition, we have now re-trained and re-evaluated our model with the input images resized to 448x448, keeping the remainder of the process of view extraction the same. Below we provide our results on multiple datasets:
>
> |              |  Soy  | FGVC Aircraft | Stanford Cars |
> |--------------|:-----:|:-------------:|:-------------:|
> | TRD: 224x224 | 52.15 | 95.60         | 96.35         |
> | TRD: 448x448 | 52.23 | 95.62         | 96.39         |
>
> With the resized input of 448x448, we observe minor improvements in the performance of TRD.
> - Pretrained weights - We thank the reviewer for pointing this out. The ResNet50 that we use as our relation-agnostic encoder were pre-trained on ImageNet1K, following existing literature [9, 78, 88]. We provide the details of our experimental settings in Section 4.1 Experimental Settings and Datasets. If the reviewer feels that we have missed something more, we would be happy to add those details as well.
> - Performance of TransFG and FFVT - Unfortunately, neither TransFG, nor FFVT report results on FGVC Aircraft in their original papers. We found the only evaluation of these methods on FGVC Aircraft to be present in [9]. For the sake of assurance, we reevaluated both TransFG and FFVT on FGVCAircraft and were able to replicate the numbers reported in [9], which is what we also report in the paper. We additionally note that our method also surpasses both TransFG and FFVT by significant margins on all other datasets as well (not just FGVC Aircraft), especially the small scale ones. We conjecture that both FFVT and TransFG being transformer-based models, are not able to cope very well with the small dataset sizes and low sample diversity of FGVC Aircraft (as well as the small scale datasets), leading to relatively lower accuracies.
>
> [a] Where to Focus: Investigating Hierarchical Attention Relationship for Fine-Grained Visual Classification, ECCV22 \
> [b] Weakly Supervised Posture Mining for Fine-Grained Classification, CVPR23 \
> [c] SR-GNN: Spatial Relation Aware Graph Neural Network for Fine-Grained Image Categorization, TIP

---

> > ### Comment · Reviewer_VJoJ · 2023-08-17
> > **Response to the rebuttal**
> >
> > Thanks for the detailed rebuttal, my concerns are properly addressed.

---

### Official Review · Reviewer_7SVw · 2023-07-07

**Soundness:** 3 good
**Presentation:** 3 good
**Contribution:** 3 good
**Rating:** 5
**Confidence:** 4

**Summary:**

The authors propose TRD, an algorithm that decomposes both input images and output classes into graphs over views by recovering transitive cross-view relationships for fine-grained visual categorization.

**Strengths:**

1. The paper is well written and easy to follow.
2. The proposed TRD is demonstrated both theoretically and empirically.

**Weaknesses:**

1. It seems that most of the experimental results in Table 1 are copied from the Relational Proxies paper. But the experimental setup seems different. For example, the number of local views is different.
2. As we know, deep GNNs usually suffer from over-smoothing issue, i.e., as the number of layers increases, the learned representations become nearly indistinguishable and the performance degrades significantly. The authors use an 8-layer GAT with 4 attention heads in each hidden layer. The reviewer wonders how the number of layers affects the performance of your models. Does the over-smoothing phenomenon exist?
3. The proposed method seems strongly related to Relational Proxies. Can you explain more about the relationship and difference between these two methods? Moreover, compared with Relational Proxies, the performance gains of TRD are very marginal, as can be seen from Table 1 and Figure 5.



**Questions:**

What is the efficiency/time complexity of the proposed method? The proposed TRD needs to construct multiple graphs. Considering the marginal performance gains, is TRD more efficient than Relational Proxies?

**Limitations:**

The authors discuss the limitations.

---

> ### Author Rebuttal · Authors · 2023-08-05
>
> **W1. Experimental settings** We agree that the number of local views is the only hyperparameter in which we differ from Relational Proxies, but the remaining settings are the same (Line 264). We note that our experiment on robustness (Figure 5) tests both TRD and Relational Proxies with the same number of views, and TRD can be seen to surpass Relational Proxies in the majority of the cases. Following the reviewer’s suggestion, we evaluate both Relational Proxies and TRD with the same number of local views in the normal (no explicit addition of noise) setting on the FGVC Aircraft dataset, and report our findings in the table below. TRD marginally outperforms Relational Proxies for all values of the number of local views, and exhibits a trend of scaling in accuracy with increasing number of local views. Relational Proxies, on the other hand, does not seem to benefit from increasing the number of local views, possibly due to its lack of robustness to noisy views.
>
> | # Local Views      | 8 | 16 | 32 | 64 |
> |--------------------|:-----------:|:-----------:|:-----------:|:-----------:|
> | Relational Proxies | 95.25     | 95.30     | 95.29     | 95.31     |
> | **TRD (Ours)**     | **95.27** | **95.45** | **95.52** | **95.60** |
>
> **W2. Over-smoothing** We thank the reviewer for suggesting this experiment, as it provides valuable insights into the ability of the GAT in TRD to learn the emergent relationships. We had initially evaluated using GATs of up to 16 layers, and had found the 8 layer version to be the best. Following the reviewer’s suggestion, we have evaluated TRD using GATs of up to 64 layers on FGVC Aircraft, presenting our findings below. We see that the performance does drop beyond 8. To validate whether this is due to the oversmoothing phenomenon, we measure the degree of distinguishability among the nodes by taking the average of their pairwise $L_2$ distances. The table shows that the distinguishability also decreases as we increase the number of layers, suggesting that the over-smoothing phenomenon does occur, as the reviewer had correctly speculated. Incidentally, even under the light of the above experiments, the 8-layer GAT remains an optimal choice for our problem.
>
> | GAT-Depth          |   4   |   8   |   16  |   32  |   64  |
> |--------------------|:-----:|:-----:|:-----:|:-----:|:-----:|
> | Accuracy           | 95.05 | **95.60** | 95.32 | 94.78 | 94.20 |
> | Distinguishability |  0.87 |  0.63 |  0.49 |  0.21 |  0.09 |
>
> **W3. Relational Proxies** The similarity between TRD (proposed) and Relational Proxies is in that they both leverage local-to-global emergent relationships to achieve the sufficiency criterion (Appendix A.3). The main difference lies in the central objectives of the two works. Relational Proxies aims to achieve SoTA performance in FGVC by learning abstract relational representations of local-to-global emergence. On the other hand, we aim to make the process of relational representation learning transparent and interpretable, by performing all computations in terms of graphs representing the learned relationships, while maintaining their performance.\
> The trade-off between performance and interpretability is a well-known phenomenon in the literature [29, 67, 21, 35]. However, TRD is not only able to retain the performance of the existing SoTA Relational Proxies, but provide marginal gains as well. This can be attributed to the (provable) robustness of TRD to noisy views. Whatever performance degradation comes from the decomposition of the abstract latent representations into graphs over image views, is recovered by the intrinsic robustness of the transitivity recovery objective. This can be seen in action in Rows 1 - 3 in Table 2 of our Ablation Studies.\
> To summarize, **our primary objective is not to surpass SoTA in FGVC, but to provide interpretability to existing SoTA algorithms that leverage relational information to achieve maximal expressivity, while retaining their performance**. The degradation in performance that comes from interpretability is compensated for by the inherent robustness of TRD, allowing us to achieve marginal performance gains over Relational Proxies, even though classification SoTA advancement is not our main objective.\
> Apart from our theoretical analyses (Sections 3.1 and 3.2) establishing the equivalence of abstract relational representation learning algorithms and our proposed TRD, we perform empirical evaluations in the Supplementary Section 3 to extract the instance-level relationships learned by Relational Proxies via a post-hoc explainer. Via TRD, we can obtain such relationships in an ante-hoc manner directly as part of the inference pipeline, without the need for any post-hoc explainers, and not only at the level of the instance, but also for the class.
>
> **Q. Compute Cost** Below we provide the computational costs of Relational Proxies and TRD in terms of wall clock time evaluated on FGVC Aircraft (same experimental settings including  # local views):
>
> |                    | Average Inference Time (ms) | Training Time (hrs) |
> |--------------------|:---------------------------:|:-------------------:|
> | Relational Proxies |             130             |          22         |
> | **TRD (Ours)**     |           **110**           |        **15**       |
>
> We can see that TRD is significantly more efficient than Relational Proxies in terms of both single sample inference as well as training time until convergence. This is because of the following reasons:
> - The Complementarity Graph in TRD is constructed exactly once before training, and the semantic relevance graph, as well as the proxy graph are learned as part of the training process.
> - TRD does not involve updating the relation-agnostic encoder $f$, which is a ResNet50, as part of the training process. Relational Proxies requires it to be updated, thereby exhibiting computationally heavier forward (as local view embeddings cannot be pre-computed) and backward passes.

---

> > ### Comment · Area_Chair_MN1w · 2023-08-21
> > **Could you please read the rebuttal and share your thoughts at your earliest convenience?**
> >
> > Cheers,
> >
> > AC

---

### Official Review · Reviewer_mGG3 · 2023-07-17

**Soundness:** 4 excellent
**Presentation:** 2 fair
**Contribution:** 3 good
**Rating:** 7
**Confidence:** 4

**Summary:**

The paper presents a novel perspective on interpretable representation learning, introducing Transitivity Recovering Decompositions (TRD) as a method for identifying graphs that can learn local-to-global representations. The proposed approach achieves state-of-the-art (SOTA) performance on Fine-Grained Visual Classification (FGVC) datasets while maintaining interpretability. The TRD is well-defined, supported by theoretical and empirical analysis, and conducts thorough interpretability and robustness experiments.

**Strengths:**

1.	The authors provide a well-defined and theoretically supported Transitivity Recovering Decompositions (TRD) method, which is further validated through empirical analysis.
2.	The experiments conducted on FGVC benchmarks demonstrate consistent SOTA performance across multiple datasets, although the improvements are marginal.
3.	The paper includes comprehensive interpretability and robustness experiments, effectively showcasing the effectiveness of TRD to a certain extent.


**Weaknesses:**

1.	The introduction section requires improvement in terms of providing a high-level overview of the proposed TRD. A simple end-to-end pipeline overview in the introduction would greatly benefit readers' understanding.
2.	The inference pipeline of the proposed system is not clearly described. Including the training and testing pseudo code would substantially enhance the clarity of the paper.
3.	The robustness analysis is limited. To comprehensively evaluate the interpretability of TRD, it is crucial to observe the results under the influence of causal interventions. For instance, replacing a percentage of local views from another class and observing the results would provide valuable insights.


**Questions:**

1.	What if there are multiple high-level concepts along with low-level concepts present in a training distribution? A quick analysis of datasets such as CIFAR100, or (if possible) ImageNet can put some light on this. Like, multiple breeds of dogs, cats, and birds are in a single classification task.
2.	How can the presence of a different local view (belonging to another class) affect the forming of cliques? (Any qualitative example or quantitative number can help understand the potential impact of TRD)
3.	What are the values of empirically defined hyper-parameters (such as delta and gamma) across the datasets? A plot/table showing the effect of such variables across the datasets is important to measure the stability of the TRD.


**Limitations:**

The authors have adequately discussed the limitations and potential positive societal impact of their work. No further discussion is necessary in this regard.

---

> ### Author Rebuttal · Authors · 2023-08-06
>
> **W1. Overview** We thank the reviewer for pointing this out. We will add the following to the final version:
> "After decomposing an input image into its constituent views following related literature [9, 78, 88], we initialize the relational representation by forming a graph through connecting complementary sets of nodes. This allows information to flow across disjoint localities, while disregarding redundancy. We then propagate this initial image graph through a trainable GNN. We obtain the class proxy graph by an online clustering of the instance node and edge embeddings. We train the GNN to match the instance and the proxy graphs by recovering transitive relationships. Concretely, this is achieved by minimizing the edit cost between the two, approximated by a learnable form of the Hausdorff Edit Distance."
>
> **W2. Pseudocode** Below, we provide PyTorch style training and inference pseudocodes for TRD. We will release our full implementation upon acceptance.\
> **Preprocess**
> ```
> def get_graphloader(X, Y, mode='train'):
> 	labels_to_instances = {}
> 	graphs = []
> 	# Complementarity graphs
> 	for x, y in dataloader(X, Y):
> 		Z_l = [local_views(x)] + [global_view]
> 		Z_v = [f(z) for z in Z]
> 		G_c = {nodes: Z_v, edges: [(z_i, z_j, repeat(1 / dot(z_i, z_j), n)) for z_j in Z_l for z_i in Z_l]}
> 		# Adding global view and its edges
> 		G_c.nodes += z_g
> 		G_c.edges += [(z_g, z_l) ones(n) for z_l in Z_l]
> 		graphs.append(G_c)
> 		if mode == 'train':
> 			labels_to_instances[y] += G_c
> 	if mode == 'test':
> 		return dataloader(graphs)
>
> 	# Initialize proxy graphs
> 	for label in Y:
> 		e = random.choice(labels_to_instances[label].all_edges())
> 		n = unique(e[0] + e[1])
> 		# Instance embeddings are only used as initializations. Proxies must be independent entities.
> 		Y_proxies[label].nodes = deepcopy(n)
> 		Y_proxies[label].edges = deepcopy(e)
> 		labels_to_instances[label] = Y_proxies
>
> 	return dataloader(labels_to_instances)
> ```
> **Training**
> ```
> for G_c, P in get_graphloader(X, Y):
> 	G_s = phi(G_c)  # semantic relevance graph
>
> 	# Assignment of instances to proxies
> 	scores = pairwise_hausdorff(G_s, P)  # [59]
> 	preds = sinkhorn(scores)  # [7]
> 	probs = softmax(preds / temp)
>
> 	loss = proxy_anchor(probs, y)  # [36]
>
> 	update(phi.params)  # Update GNN
> 	update(P)  # Update proxy centroids
> ```
> **Inference**
> ```
> # X: Test images; P: Trained Class Proxies
>
> for G_c in get_graphloader(X, None, mode='test'):
> 	G_s = phi(G_c)  # semantic relevance graph
> 	# Assignment of instances to proxies
> 	scores = pairwise_hausdorff(G_s, P)  # [59]
> 	pred = argmax(scores,  dim=1)
> ```
> **W3 and Q2. Causality** We thank the reviewer for suggesting these insightful experiments on robustness. To this end, we train TRD by replacing a subset of the local views for each instance with local views from other classes, both during training and inference. As the proxies are obtained via a clustering of the instance graphs, these local views consequently influence the proxy graphs. We report our quantitative and qualitative findings in the **pdf attached as part of the global response.**\
> TRD significantly outperforms Relational Proxies [9] at all noise rates, and the gap between their performances widens as the percentage of corruption increases (Tab 1, attached pdf).
> Qualitatively (Fig 1, attached pdf), our model successfully disregards the views introduced from the negative class at both the instance and proxy level. Such views can be seen as being very weakly connected to the global view, as well as the correct set of local views that actually belong to that class.
> **Under this causal intervention, the TRD objective is thus equivalent to performing classification while having access to only the subgraph of clean views from the correct class.**
>
> **Q1. Coarse-grained** Following the reviewer’s suggestion, and existing FGVC literature [9, 19], we evaluate the contribution of our novel Transitivity Recovery objective in the coarse-grained (multiple fine-grained subcategories in a single class) and fine-grained subsets of ImageNet, namely Tiny ImageNet and Dogs ImageNet (Stanford Dogs) and report our findings below. Although our method can surpass existing SoTA in both the settings, larger gains ($\Delta$) are achieved in the fine-grained setting, suggesting that TRD is particularly well suited for that purpose.
>
> |                            | Tiny ImageNet | Dogs ImageNet |
> |----------------------------|:-------------:|:-------------:|
> | MaxEnt [19]  |     82.29     |     75.66     |
> | Relational Proxies [9]  |     88.91     |     92.75     |
> | $a =$ w/o Transitivity Recovery  |     88.10     |     91.03     |
> | $b =$ with Transitivity Recovery |     89.02     |     93.10     |
> | $\Delta = (b - a)$                      |      0.92     |    **2.07**   |
>
> **Q3. Bounds** Delta and gamma are not hyperparameters, but intrinsic properties of the dataset, and as such, cannot be controlled explicitly. Respectively, they denote implicit distance and mutual information bounds that the learning process aims to optimize for. We leverage them to prove the theoretical equivalence of existing abstract relational representation learning algorithms and our interpretable version.
> Specifically, delta is an estimate of the best achievable error (Bayes error rate). Gamma is a way of quantifying the amount of emergence encoded by pairs of views, i.e., how important it is to jointly (rather than  individually) observe the two views in order to determine the global structure of the object.\
> One implicit way of controlling both delta and gamma at the level of the dataset is by altering the set of local views. We do this as part of addressing W3 and Q2, by introducing local views from other classes at both the instance and the proxy level, as well as our experiment on Robustness to Noisy Views in the main manuscript. The results show that TRD is able to efficiently optimize for both of these parameters by identifying the most relevant subgraph comprising local views of the target class.

---

> > ### Comment · Reviewer_mGG3 · 2023-08-11
> > **Rebuttal response**
> >
> > I appreciate the response and the thorough additional experiments conducted by the authors. These efforts have considerably clarified several of my uncertainties. Particularly, the incorporation of new graph visualizations involving causal interventions provides substantial support for the paper's central proposition.
> >
> > For optimal clarity and comprehension, I recommend that the authors consider enhancing their presentation in accordance with the suggestions outlined. I am inclined to elevate my rating, keeping in mind the anticipation that the authors will adequately substantiate the rationale behind the new experiments in the forthcoming revised version, should it be accepted.

---

> > > ### Author Response · Authors · 2023-08-14
> > > **Note of thanks**
> > >
> > > We thank the reviewer for going through our rebuttal response, validating the findings, and increasing their score. We will update the final version with all the new experimental results along with their underlying rationale, as well as the clarifications that we have provided as part of the rebuttal.

---

### Author Rebuttal · Authors · 2023-08-06

We thank the reviewers for their valuable comments and feedback. We have addressed their individual concerns in their respective rebuttal sections. Here we attach some qualitative results for addressing the comment by Reviewer mGG3 on experimenting with causal interventions.

N.B. It can be observed that we choose FGVC Aircraft for performing some of the experiments requested by the reviewers. Our choice was motivated by the fact that the performance of our model on FGVC Aircraft is generally reflective of the general trends across other datasets, because of its challenging low intra-class and high inter-class similarities. Also, the size of the dataset is reasonable enough for us to complete all the experiments suggested by the reviewers within the rebuttal time frame.

---

### Decision · Program_Chairs · 2023-09-21

**Decision:**

Accept (poster)

**Comment:**

The draft presents a well-defined and theoretically supported Transitivity Recovering Decompositions (TRD) method, with both empirical analysis and comprehensive experiments. The empirical results demonstrate consistent state-of-the-art (SOTA) performance across various Fine-Grained Visual Categorization (FGVC) benchmarks. The writing is clear and coherent, as pointed out by all reviewers. Given the strong motivation, solid technique, and achieved SOTA performance on multiple FGVC benchmarks, along with well-conducted ablation studies and visualization, I recommend accepting this paper. I urge the authors consider to enhance their presentation in accordance with the suggestions outlined by reviewers.